# Identification and Estimation of Treatment Effects under Coupled Confounding and Collider Biases

## Abstract

In causal inference, confounding bias and collider bias pose two major challenges for treatment effect estimation in observational studies. Confounding bias arises from unobserved common factors that simultaneously affect the treatment and outcome, while collider bias results from non-random sample selection caused by both variables. Existing methods focus on bias correction for a specific bias, such as using Instrumental Variables (IVs) to address confounding bias and Selection IVs (SIVs) to mitigate collider bias. However, real-world data frequently exhibit coupled confounding and collider biases, where unmeasured confounders directly affect the selection mechanism. Currently, the coupled biases problem remains an unaddressed challenge. In this paper, we propose a new identification theory for treatment effects under coupled biases with an IV set, which contains subsets serving as IV and SIV, respectively. Based on this theory, we propose a novel treatment effect estimation method, DualDebiasIV (DDIV), which decomposes the IV set to separately obtain the SIV and IV, using them for biases decoupling and correction. To the best of our knowledge, this is the first work to provide a solution for the identification and estimation of treatment effects under coupled biases. DDIV is theoretically guaranteed, with proofs provided for the correctness of the decomposition and the consistency of the estimates. Extensive experimental results on semi-synthetic and real-world datasets show that DDIV achieves significant performance improvements, further demonstrating its practical effectiveness.

## 1 Introduction

Causal inference has made significant progress in statistics, medicine, and economics (D'Agostino, 2007; Varian, 2016). With the advent of the machine learning boom, causal inference also contributes to improving the interpretability, stability, and fairness of machine learning models (Pearl, 2009; Cui and Athey, 2022). The gold standard for causal inference is conducting Randomized Controlled Trials (RCTs) to estimate treatment effects (Imbens and Rubin, 2015). However, in real-world scenarios, RCTs are often expensive or even impracti-

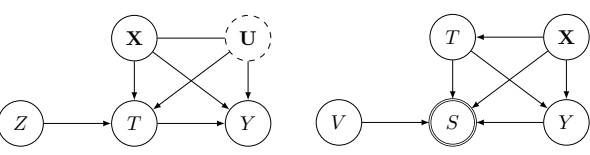

(a) Confounding bias and IV.  (b) Collider bias and SIV.

Figure 1: Causal graphs illustrating various biases and their solutions. Dashed nodes represent unobserved variables, undirected links denotes correlations between variables, and double-lined nodes represent controlled variables.

cal (Rosner, 2003). Therefore, the core task of causal inference is to estimate treatment effects from observational data. However, observational data may be subject to multiple biases, primarily due to confounding bias and collider bias (Ellenberg, 1994; Munafò et al., 2018; Hernán and Robins, 2020).

Let $\mathbf{U}$ denote unmeasured confounders, $\mathbf{X}$ denote observed covariates, $Z$ denote the Instrumental Variable (IV), $V$ denote the Selection IV (SIV), $T$ denote the treatment, $Y$ denote the outcome, and $S$ denote the selection indicator. $S$ equals 1 if $Y$ is observed, meaning the sample is selected into

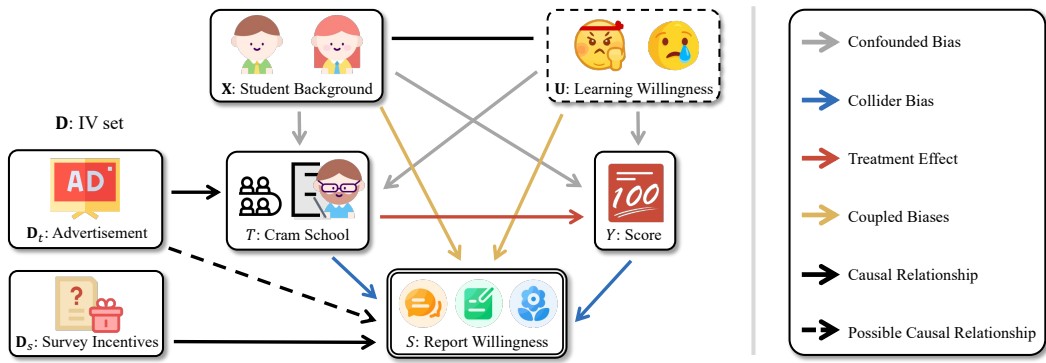

Figure 2: A real-world example illustrating the problem and motivation.

the observations, and 0 otherwise. Confounding bias arises when unobserved variables influence both the treatment and the outcome. As shown in Figure 1(a), the backdoor path $T \leftarrow \mathbf{U} \rightarrow Y$ induces confounding bias, which cannot be blocked through adjustment because $\mathbf{U}$ is unobserved. IVs, such as $Z$ in Figure 1(a), can address this bias via two-stage regression (Angrist et al., 1996). Selection bias arises when the observations are not representative of the target population due to non-random sample selection. A typical form of selection bias is collider bias, which occurs when the selection mechanism is influenced by both $T$ and $Y$, as shown by the path $T \rightarrow S \leftarrow Y$ in Figure 1(b). Conditioning on $S = 1$ opens this path and induces bias. SIVs, such as $V$ in Figure 1(b), are used to mitigate this bias (Miao et al., 2024). Existing methods, such as IV and SIV, are designed to handle either confounding or collider bias but cannot address both when they are coupled, i.e., when unobserved confounders also affect sample selection ($\mathbf{U} \rightarrow S$).

In real-world scenarios, observational data frequently exhibit coupled confounding and collider biases (Greenland, 2003; Hernán and Robins, 2020). For example, as illustrated in Figure 2, in evaluating the impact of cram school attendance ($T$) on exam grades ($Y$), $S$ refers to whether a student is willing to report their grades, and observed covariates $\mathbf{X}$ include background factors. Unobserved factors like learning willingness ($\mathbf{U}$) influence both the likelihood of attending cram school ($T$) and exam grades ($Y$), inducing confounding bias. Meanwhile, students who both attend cram school ($T$) and perform well ($Y$) are more likely to report their grades ($S$), leading to collider bias ($T \rightarrow S \leftarrow Y$). Learning willingness ($\mathbf{U}$) can also affect their report willingness ($\mathbf{U} \rightarrow S$), introducing coupled biases. However, to the best of our knowledge, the identification and estimation of treatment effects under coupled biases remain an unresolved challenge.

To address the challenge, we propose a new identification theory for treatment effects under coupled biases, demonstrating the identifiability of treatment effects under the assumption that an IV set satisfying the following conditions is known: First, the IV set satisfies the standard exclusion and unconfounded conditions (Angrist et al., 1996; Hartford et al., 2017); and second, subsets within this set are relevant to $T$ and $S$, respectively. In other words, the IV set contains subsets serving as IV and SIV, respectively. To aid in understanding this assumption, we use the cram school example to illustrate the IV set. As shown in Figure 2, the IV set ($\mathbf{D}$) in this example includes encouragement to attend the cram school ($\mathbf{D}_t$), such as advertisements, and encouragement to report exam grades ($\mathbf{D}_s$), such as survey incentives. They are both unrelated to individual learning willingness (unconfounded) and do not directly influence exam performance (exclusion). The former is relevant to participation in the school ($\mathbf{D}_t \rightarrow T$), while the latter is relevant to the willingness to report exam grades ($\mathbf{D}_s \rightarrow S$) and is unrelated to school participation ($\mathbf{D}_s \nrightarrow T$).

Based on the developed identification theory, we propose a novel two-stage treatment effect estimation method, DualDebiasIV (DDIV), which decomposes the IV set to separately obtain the SIV and IV, using them for bias decoupling and correction. In the first stage, DDIV decomposes the representations satisfying the IV and SIV conditions from the IV sets, respectively, using mutual information constraints. Simultaneously, it performs regression on the treatment using the IV representation and estimates the selection score using the SIV representation. In the second stage, DDIV reweights the sample using the inverse of the selection score and regresses the outcome on the SIV representation, the estimated treatment, and the observed covariates, yielding unbiased treatment effect estimation.

We provide theoretical proofs for the correctness of the decomposed representations and the consistency of the treatment effect estimates. Extensive experimental results on both semi-synthetic and real-world datasets show that DDIV achieves significant performance improvements, demonstrating its effectiveness. In summary, the contributions of this paper are as follows:

- We investigate an important problem in causal inference, i.e., coupled confounding and collider biases, which remains an unresolved challenge.
- We propose a novel identification theory for treatment effects under coupled biases. To the best of our knowledge, this is the first work to prove that treatment effects are identifiable with an IV set under coupled biases.
- We propose a novel method with theoretical guarantees, namely DDIV, which achieves an unbiased estimate of treatment effects under coupled biases. To the best of our knowledge, it is the first method that can address this challenge.
- We conducted extensive experimental validation of DDIV, using both semi-synthetic and real datasets. The results show that DDIV achieves significant performance improvements.

## 2 PRELIMINARIES

### 2.1 PROBLEM SETUP

The observational dataset $\mathbb{D} = \{\mathbf{x}_i, t_i, y_i, \mathbf{d}_i, s_i\}_{i=1}^{n}$ is sampled from the target population $\mathcal{P}$. For each sample $i$ in the dataset, $\mathbf{x}_i$ denotes the covariate vector, $t_i$ denotes the received treatment, $\mathbf{d}_i$ denotes the IV set vector, $y_i$ denotes the outcome, and $s_i$ denotes the selection indicator, indicating whether the outcome is observed: $s_i = 1$ means $y_i$ is observable, while $s_i = 0$ means $y_i$ is missing and unobservable. The objective is to estimate $\mathbb{E}[Y \mid do(T = t), \mathbf{X}]$. However, due to the presence of coupled confounding and collider biases, the counterfactual outcomes and the outcome values of the $S = 0$ samples are not identifiable. Therefore, how to estimate $\mathbb{E}[Y \mid do(T = t), \mathbf{X}]$ under coupled biases is the problem this paper seeks to address. Following previous research, we establish the following assumptions throughout this paper (Imbens and Rubin, 2015):

**Assumption 2.1. Stable Unit Treatment Value Assumption.** The potential outcome distribution of a unit does not depend on the treatment assignment of another unit.

**Assumption 2.2. Overlap Assumption.** Every unit possesses a nonzero likelihood of receiving treatment and being selected, i.e., $0 < \mathbb{P}(T = 1 \mid \mathbf{X} = \mathbf{x}) < 1$ and $0 < \mathbb{P}(S = 1 \mid \mathbf{X} = \mathbf{x}) < 1$.

### 2.2 INSTRUMENTAL VARIABLE

The Instrumental Variable (IV) serves as effective tools for addressing confounding bias and is defined as follows (Angrist et al., 1996; Hernán and Robins, 2006; Hartwig et al., 2023).

**Definition 2.3. Instrumental Variable.** An instrumental variable $Z$ satisfies: (1) **Relevance**: $Z$ is relevant to $T$, i.e., $Z \not\perp\!\!\!\perp T \mid \mathbf{X}$. (2) **Exclusion**: $Z$ does not directly affect $Y$, i.e., $Z \perp\!\!\!\perp Y \mid \mathbf{U}, \mathbf{X}, T$. (3) **Unconfounded**: $Z$ is independent of all confounders, including $\mathbf{X}$ and $\mathbf{U}$, i.e., $Z \perp\!\!\!\perp \mathbf{U}, \mathbf{X}$.

**Assumption 2.4. Homogeneous Treatment-Outcome Association.** The association between the treatment and outcome is homogeneous in the different levels of unmeasured confounders, i.e., $\mathbb{E}[Y \mid T = a, \mathbf{U}] - \mathbb{E}[Y \mid T = b, \mathbf{U}] = \mathbb{E}[Y \mid T = a] - \mathbb{E}[Y \mid T = b]$.

Under Assumption 2.4 (Heckman et al., 2006; Hernán and Robins, 2006; Hartwig et al., 2023), $\mathbb{E}[Y \mid do(T = t), \mathbf{X}]$ can be identified with IVs, and we can obtain its unbiased estimate through two-stage regression if collider bias is absent. However, once there are coupled confounding and collider biases, $Z$ is correlated with $\mathbf{U}$ through the open path $Z \to T \to S \leftarrow \mathbf{U}$ conditional on $S = 1$, resulting in the IV no longer satisfying the unconfounded condition.

### 2.3 SELECTION INSTRUMENTAL VARIABLE

The Selection IV (SIV) is used to address collider bias in the absence of unobserved confounders, with the following definition (Heckman, 1979; 1997; Sun et al., 2018).

**Definition 2.5. Selection Instrumental Variable.** A valid selection instrumental variable $V$ is supposed to satisfy the following conditions: (1) **Exclusion**: $V$ does not directly affect the outcome, i.e., $V \perp\!\!\!\perp Y \mid \mathbf{X}, T$. (2) **Relevance**: $V$ is conditionally correlated with $S$, i.e., $V \not\perp\!\!\!\perp S \mid \mathbf{X}, T$.

Definition 2.5 indicates that the SIV is a direct cause of the sample selection but not of the outcome, which can help identify $\mathbb{P}(\mathbf{X}, T, Y, V, S)$ using the selection score, with the following definition.

**Definition 2.6. Selection Score.** The selection score is the conditional probability of the selection mechanism given all observed variables. Formally, it is expressed as:

$$\pi(\mathbf{X}, T, Y, V) = \mathbb{P}(S = 1 \mid \mathbf{X}, T, V, Y),$$

With the help of the selection score, the SIV theory establishes the sufficient conditions for the identification of $\mathbb{E}[Y \mid do(T = t), \mathbf{X}]$ as follows (Sun et al., 2018).

**Condition 2.7.** $\forall \theta_1 \neq \theta_2$, the ratio $\frac{\pi(\mathbf{X}, T, Y, V; \theta_1)}{\pi(\mathbf{X}, T, Y, V; \theta_2)}$ is either a constant or varies with $\mathbf{X}$, $T$, and $V$, where $\pi(\mathbf{X}, T, Y, V; \theta_i)$ denotes a candidate for $\pi(\mathbf{X}, T, Y, V)$ indexed by the parameters $\theta_i$.

**Theorem 2.8.** *The joint distribution $\mathbb{P}(\mathbf{X}, T, Y, V, S)$ is identifiable if Condition 2.7 holds.*

Theorem 2.8 guarantees the identifiability of the joint distribution of the full data. Therefore, under the assumption of no unobserved confounders, $\mathbb{E}[Y \mid do(T = t), \mathbf{X}]$ is also identifiable. The proof of Theorem 2.8 has been established in previous studies (Sun et al., 2018), primarily based on contradiction, and we provide the full proof in Appendix B.1. To ensure the satisfaction of Condition 2.7 within the observational data, in addition to the assumption of no unobserved confounders, previous studies also make the following assumption (Sun et al., 2018).

**Assumption 2.9. Separable Selection Model Assumption.** The selection score follows an additively separable model, i.e., $\text{logit}(\mathbb{P}(S = 1 \mid \mathbf{X}, T, V, Y)) = q(\mathbf{X}, T, V) + h(\mathbf{X}, T, Y)$, where $q$ and $h$ are nonlinear functions.

Assumption 2.9 imposes an additive separability constraint on the effects of $Y$ and $V$ on $S$, ensuring that Condition 2.7 is satisfied. With these theoretical results, one can estimate $\mathbb{E}[Y \mid do(T = t), \mathbf{X}]$ under collider bias using methods such as Inverse Probability Weighting (IPW) (Sun et al., 2018). However, in the presence of coupled biases, the existing SIV theory is insufficient to ensure the satisfaction of Condition 2.7 and the identification of $\mathbb{E}[Y \mid do(T = t), \mathbf{X}]$.

## 3 THEORY

### 3.1 IDENTIFICATION

As discussed earlier, both IV and SIV methods fail to guarantee the identification of treatment effects in the presence of coupled biases. Therefore, we propose a novel theory for identifying treatment effects under coupled biases, which requires two key assumptions.

**Assumption 3.1. Known Instrumental Variable Set Assumption.** There exists a known set of variables, denoted as $\mathbf{D}$, which satisfies the following conditions: (1) **Exclusion**: $\mathbf{D}$ does not directly affect the outcome $Y$, i.e., $\mathbf{D} \perp\!\!\!\perp Y \mid \mathbf{U}, \mathbf{X}, T$. (2) **Unconfounded**: $\mathbf{D}$ is independent of all confounders, including $\mathbf{X}$ and $\mathbf{U}$, i.e., $\mathbf{D} \perp\!\!\!\perp \mathbf{U}, \mathbf{X}$. (3) **Separable subset**: There exists two subsets, $\mathbf{D}_t, \mathbf{D}_s \subset \mathbf{D}$, such that $\mathbf{D}_t \not\perp\!\!\!\perp T \mid \mathbf{X}$, $\mathbf{D}_s \not\perp\!\!\!\perp S \mid \mathbf{X}, T$, and $\mathbf{D}_s \perp\!\!\!\perp T$.

Assumption 3.1 indicates the existence of subsets in the IV set that satisfy the conditions of Definition 2.3 and Definition 2.5, respectively. Specifically, $\mathbf{D}_t$ satisfies Definition 2.3, while $\mathbf{D}_s$, due to its independence from all variables except $S$, satisfies Definition 2.5. This assumption is often satisfied in scenarios where IVs are available, as any valid IV is by definition $\mathbf{D}_t$, while some weak IVs, defined as those with weak to almost no correlation with $T$ (Andrews et al., 2019), are considered as $\mathbf{D}_s$ if they also influence sample selection. Moreover, in real-world applications, we can use sensitivity analysis to assess the validity of Assumption 3.1(1) and (2) (Baiocchi et al., 2014). Then, Assumption 3.1(3) can also be verified by directly calculating the correlation coefficients between each IV and $T$, and between each IV and $S$. That is, if an IV is correlated with $T$, it belongs to $\mathbf{D}_t$; if an IV is independent of $T$ and correlated with $S$, it belongs to $\mathbf{D}_s$. We also provide a real-world example that satisfies this assumption, as illustrated in Figure 2.

**Assumption 3.2. Separable Selection Model with Unobserved Factors Assumption.** The selection score follows an additively separable model, i.e., $\text{logit}(\mathbb{P}(S = 1 \mid \mathbf{U}, \mathbf{X}, T, \mathbf{D}_s, Y)) = q(\mathbf{X}, T, \mathbf{D}_s) + h(\mathbf{U}, \mathbf{X}, T, Y)$, where $q$ and $h$ are nonlinear functions.

Assumption 3.2 is a relaxation of Assumption 2.9. Assumption 2.9 does not allow $\mathbf{U}$ to affect $S$. In contrast, Assumption 3.2 allows an additive separability constraint on the effects of $\mathbf{U}, Y$, and $\mathbf{D}_s$ on $S$. Based on the above assumptions, we propose the identification theory of $\mathbb{E}[Y \mid do(T = t), \mathbf{X}]$ under coupled biases.

**Theorem 3.3.** *Under Assumptions 2.1, 2.2, 2.4, 3.1, and 3.2, $\mathbb{E}[Y \mid do(T = t), \mathbf{X}]$ is identifiable under coupled confounding and collider biases.*

*Proof.* To prove that $\mathbb{E}[Y \mid do(T = t), \mathbf{X}]$ is identifiable, we first prove that $\mathbb{P}(\mathbf{X}, T, Y, \mathbf{D}_s)$ is identifiable, then demonstrate that $\mathbb{E}[Y \mid do(T = t), \mathbf{X}, \mathbf{D}_s]$ can be identified from $\mathbb{P}(\mathbf{X}, T, Y, \mathbf{D}_s)$ with the help of $\mathbf{D}_t$, and finally, $\mathbb{E}[Y \mid do(T = t), \mathbf{X}]$ is identified based on $\mathbb{E}[Y \mid do(T = t), \mathbf{X}, \mathbf{D}_s]$ and $\mathbb{P}(\mathbf{D}_s)$. The detailed proof is provided in Appendix B.2. $\square$

Note that while the proof of Theorem 3.3 builds upon IV and SIV theories, it is not simply a straightforward combination of the two. A simple combination is insufficient to establish the identification of treatment effects under coupled biases, as their key assumptions are not satisfied.

## 3.2 ESTIMATION

Based on the identification theory, the estimation of $\mathbb{E}[Y \mid do(T = t), \mathbf{X}]$ lies in the estimation of $\mathbb{E}[Y \mid do(T = t), \mathbf{X}, \mathbf{D}_s]$. Therefore, we propose an estimation procedure for it under coupled biases with the IV set, consisting of two estimators. The first estimator aims to estimate the selection score using $\mathbf{D}_s$. The second estimator aims to estimate $\mathbb{E}[Y \mid do(T = t), \mathbf{X}, \mathbf{D}_s]$ with the help of the selection score and $\mathbf{D}_t$, detailed below.

**Selection score estimator.** Following previous work (Sun et al., 2018), we decompose the estimation of the selection score into two parts:

$$\pi(\mathbf{X}, T, Y, \mathbf{D}_s) = (1 + \exp(-\rho(\mathbf{X}, T, \mathbf{D}_s) - \eta(\mathbf{X}, T, Y, \mathbf{D}_s)))^{-1}$$

Here, $\rho(\mathbf{X}, T, \mathbf{D}_s)$ is the baseline conditional odds of observing the complete data of a unit:

$$\rho(\mathbf{X}, T, \mathbf{D}_s) = \log\left(\frac{\mathbb{P}(S = 1 \mid \mathbf{X}, T, Y = 0, \mathbf{D}_s)}{\mathbb{P}(S = 0 \mid \mathbf{X}, T, Y = 0, \mathbf{D}_s)}\right),$$

where $Y = 0$ is a reference value, which can be other values within the value range of $Y$.

$\eta(\mathbf{X}, T, Y, \Phi_s)$ is the collider bias function defined as:

$$\eta(\mathbf{X}, T, Y, \mathbf{D}_s) = \log\left(\frac{\mathbb{P}(S = 1 \mid \mathbf{X}, T, Y, \mathbf{D}_s)/\mathbb{P}(S = 0 \mid \mathbf{X}, T, Y, \mathbf{D}_s)}{\mathbb{P}(S = 1 \mid \mathbf{X}, T, Y = 0, \mathbf{D}_s)/\mathbb{P}(S = 0 \mid \mathbf{X}, T, Y = 0, \mathbf{D}_s)}\right)$$

The collider bias function quantifies the degree of association between $Y$ and $S$ given $\mathbf{X}, T$, and $\mathbf{D}_s$ on the log odds ratio scale, which captures the effect of $Y$ on $S$.

Based on the above decomposition, the estimation of the selection score is reduced to the estimation of $\rho(\mathbf{X}, T, \mathbf{D}_s)$ and $\eta(\mathbf{X}, T, Y, \mathbf{D}_s)$, which can be achieved by optimizing the following objective.

$$\hat{\zeta}, \hat{\omega} = \arg\min_{\zeta, \omega} \|\mathbb{E}[\psi(\mathbf{X}, T, \mathbf{D}_s; \zeta, \omega)]\|^2 + \|\mathbb{E}[\delta(\mathbf{X}, T, Y, \mathbf{D}_s; \zeta, \omega)]\|^2.$$

Here,

$$\psi(\mathbf{X}, T, \mathbf{D}_s; \zeta, \omega) = \left(S \cdot \pi(\mathbf{X}, T, Y, \mathbf{D}_s; \zeta, \omega)^{-1} - 1\right) \cdot h_1(\mathbf{X}, T, \mathbf{D}_s), \tag{1}$$

$$\delta(\mathbf{X}, T, Y, \mathbf{D}_s; \zeta, \omega) = \frac{(h_2(\mathbf{X}, T, \mathbf{D}_s) - \mathbb{E}[h_2(\mathbf{X}, T, \mathbf{D}_s) \mid \mathbf{X}, T; \hat{v}]) \cdot S \cdot g(\mathbf{X}, T, Y)}{\pi(\mathbf{X}, T, Y, \mathbf{D}_s; \zeta, \omega)}, \tag{2}$$

where $\zeta$ denotes candidate parameters for $\mathbb{P}(S \mid \mathbf{X}, T, Y = 0, \mathbf{D}_s)$, $\omega$ denotes candidate parameters for $\eta(\mathbf{X}, T, Y, \mathbf{D}_s)$, $h_1(\mathbf{X}, T, \mathbf{D}_s)$ denotes the basis function of $\mathbb{P}(S \mid \mathbf{X}, T, Y = 0, \mathbf{D}_s)$, $g(\mathbf{X}, T, Y)$ and $h_2(\mathbf{X}, T, \mathbf{D}_s)$ denote the basis functions of $\eta(\mathbf{X}, T, Y, \mathbf{D}_s)$, $\hat{\zeta}$ denotes the estimated parameters

of $\mathbb{P}(S \mid \mathbf{X}, T, Y = 0, \mathbf{D}_s)$, $\hat{\omega}$ denotes the estimated parameters of $\eta(\mathbf{X}, T, Y, \mathbf{D}_s)$, and $\hat{\upsilon}$ denotes the estimated parameters of $\mathbb{E}\left[h_2(\mathbf{X}, T, \mathbf{D}_s) \mid \mathbf{X}, T\right]$, which is obtained by optimizing the following objective function.

$$\hat{\upsilon} = \arg \min_{\upsilon} \|h_2(\mathbf{X}, T, \mathbf{D}_s) - \mathbb{E}\left[h_2(\mathbf{X}, T, \mathbf{D}_s) \mid \mathbf{X}, T; \upsilon\right]\|^2,$$

where $\upsilon$ denotes candidate parameters for $\mathbb{E}\left[h_2(\mathbf{X}, T, \mathbf{D}_s) \mid \mathbf{X}, T\right]$.

**IV-based IPW estimator of the outcome.** First, similar to most IV studies (Angrist et al., 1996; Hernán and Robins, 2006; Hartwig et al., 2023), we regress $T$ on $\mathbf{X}$ and $\mathbf{D}_t$ to obtain the estimated value of $T$, denoted as $\hat{T}$. Next, we use the inverse of the selection score to reweight the samples with $S = 1$ and perform a regression of $Y$ on $\mathbf{X}$, $\hat{T}$, and $\mathbf{D}_s$ using the reweighted samples. The objective function is as below.

$$\hat{\xi} = \arg \min_{\xi} \left\| S \cdot Y \cdot \pi(\mathbf{X}, \hat{T}, Y, \mathbf{D}_s; \hat{\zeta}, \hat{\omega})^{-1} - \mathbb{E}\left[Y \mid \mathbf{X}, \hat{T}, \mathbf{D}_s; \xi\right] \right\|^2,$$

where $\xi$ denotes candidate parameters for $\mathbb{E}[Y \mid \mathbf{X}, \hat{T}, \mathbf{D}_s]$, and $\hat{\xi}$ denotes the estimated parameters of $\mathbb{E}[Y \mid \mathbf{X}, \hat{T}, \mathbf{D}_s]$.

**Theorem 3.4.** *Under Assumptions 2.1, 2.2, 2.4, 3.1, and 3.2, if $\pi(\mathbf{X}, T, Y, \mathbf{D}_s; \zeta, \omega)$, $\mathbb{E}\left[h_2(\mathbf{X}, T, \mathbf{D}_s) \mid \mathbf{X}, T; \upsilon\right]$, and $\mathbb{E}[Y \mid \mathbf{X}, \hat{T}, \mathbf{D}_s; \xi]$ are correctly specified, both the selection score estimator and the outcome estimator are consistent and asymptotically normal as $n \to \infty$.*

*Proof.* We prove the unbiasedness of the two estimators, and then the consistency and asymptotic normality of them in large samples holds under the conditions specified in previous work (Newey and McFadden, 1994; Newey and Powell, 2003; Sun et al., 2018). First, we prove the unbiasedness of the selection score estimator. Next, we prove the unbiasedness of the outcome estimator. Finally, under the conditions specified in previous work (Newey and McFadden, 1994; Newey and Powell, 2003; Sun et al., 2018), $\upsilon^*, \eta^*, \omega^*$, and $\xi^*$ are the probability limits of $\hat{\upsilon}, \hat{\eta}, \hat{\omega}$, and $\hat{\xi}$, respectively. Therefore, the consistency and asymptotic normality of the two estimators in large samples holds. The detailed proof is in Appendix B.3. □

## 4 METHODOLOGY

Based on the above theories, we propose DualDebiasIV (DDIV), a novel approach for causal inference under coupled biases. According to Assumption 3.1, the IV set $\mathbf{D}$ contains subsets that satisfy the conditions of IV and SIV, respectively. However, in practical applications, it is often unknown which elements of $\mathbf{D}$ satisfy the IV condition and which satisfy the SIV condition. Therefore, during the treatment effect estimation process, DDIV automatically learns decomposed representations, $\Phi_t$ and $\Phi_s$, from the IV set. $\Phi_t$ corresponds to the representation that satisfies the IV condition, and $\Phi_s$ corresponds to the representation that satisfies the SIV condition. Specifically, DDIV consists of the following two stages.

- **Stage 1.** DDIV decomposes the IV and SIV representations from the IV set, respectively, using mutual information constraints. Simultaneously, it performs regression on the treatment using the IV representation and estimates the selection score using the SIV representation.
- **Stage 2.** DDIV reweights the sample using the inverse of the selection score and regresses the outcome on the SIV representation, the estimated treatment, and the observed covariates, yielding unbiased treatment effect estimation.

### 4.1 THE FIRST-STAGE OF DDIV

In the first stage of DDIV, we begin by learning the decomposed representations of $D$, i.e., $\Phi_t$ and $\Phi_s$. Based on Assumption 3.1, to achieve the decomposition, we must impose the following constraints on $\Phi_t$ and $\Phi_s$: (1) $\Phi_t \not\perp T \mid \mathbf{X}$, (2) $\Phi_s \not\perp S \mid \mathbf{X}, T$, (3) $\Phi_s \perp\!\!\!\perp T$, and (4) $\Phi_t \perp\!\!\!\perp \Phi_s$. To achieve the above objectives, we learn two representation model, $\Phi_t \equiv f_{\Phi_t}(D)$ and $\Phi_s \equiv f_{\Phi_s}(D)$, by: (1) maximizing the predictive power of $\Phi_t$ on $T$, (2) maximizing the predictive power of $\Phi_s$ on $S$, (3)

minimizing the Mutual Information (MI) between $\Phi_s$ and $T$, and (4) minimizing the MI between $\Phi_t$ and $\Phi_s$. The loss function is as follows:

$$\mathcal{L}_{f_t, f_s, f_{\Phi_t}, f_{\Phi_s}} = -\frac{1}{n} \sum_{i:s_i=1} \log(f_s(\mathbf{x}_i, t_i, f_{\Phi_s}(\mathbf{d}_i))) - \frac{1}{n} \sum_{i:s_i=0} \log(1 - f_s(\mathbf{x}_i, t_i, f_{\Phi_s}(\mathbf{d}_i)))$$

$$+ \frac{1}{n} \sum_{i=1}^{n} (t_i - f_t(\mathbf{x}_i, f_{\Phi_t}(\mathbf{d}_i)))^2 + \lambda_1 \cdot \mathcal{I}(T, f_{\Phi_s}(\mathbf{D})) + \lambda_2 \cdot \mathcal{I}(f_{\Phi_s}(\mathbf{D}), f_{\Phi_t}(\mathbf{D})),$$

where $f_t$ and $f_s$ are the treatment and selection models, $\lambda_1$ and $\lambda_2$ are hyperparameters, and $\mathcal{I}$ denotes the MI between two distributions, indicating the strength of correlation between them. Note that $f_t$ and $f_s$ are simultaneously learned with $f_{\Phi_t}$ and $f_{\Phi_s}$. Here, we use Contrastive Log-ratio Upper Bound (CLUB) (Cheng et al., 2020) to achieve MI estimation and minimization. The following proposition guarantees the correctness of the decomposed representations.

**Proposition 4.1.** *Under Assumption 3.1, if the parameters of the models are sufficiently rich, the MI estimator can approximate the upper bound of mutual information, and $\lambda_1, \lambda_2$ are sufficiently large, then there exist solutions $\hat{f}_t$, $\hat{f}_s$, $\hat{f}_{\Phi_t}$, and $\hat{f}_{\Phi_s}$, such that $\hat{f}_{\Phi_t}(\mathbf{D})$ satisfies the IV conditions and $\hat{f}_{\Phi_s}(\mathbf{D})$ satisfies the SIV conditions.*

*Proof.* Under Assumption 3.1, all variables in the IV set satisfy the exclusion and unconfounded conditions for IV. Since the representations are functions of the original variables, these independence conditions are preserved. Therefore, what remains to be proven is that there exist solutions to the objective function of DDIV that satisfy the following conditions: (1) $\hat{f}_{\Phi_t}(\mathbf{D}) \not\perp\!\!\!\perp T \mid \mathbf{X}$, (2) $\hat{f}_{\Phi_s}(\mathbf{D}) \not\perp\!\!\!\perp S \mid \mathbf{X}, T$, and (3) $\hat{f}_{\Phi_s}(\mathbf{D}) \perp\!\!\!\perp Y \mid \mathbf{X}, T$, which is detailed in Appendix B.4. $\qquad\square$

Next, we can directly use the learned $\hat{f}_t$ and $\hat{f}_{\Phi_t}$ to obtain $\hat{T} = \hat{f}_t(\mathbf{X}, \hat{f}_{\Phi_t}(\mathbf{D}))$. Note that during the learning process of $f_t$ and $f_{\Phi_t}$, we use all the samples in $\mathcal{D}$ without conditional on $S$. Therefore, the estimated treatment $\hat{T}$ is not affected by collider bias. However, we cannot directly regress $Y$ on $\hat{T}$ using $S = 1$ samples, due to the open path $\hat{T} \to S \leftarrow \mathbf{U}$ conditional on $S = 1$ caused by collider bias, making $\hat{T}$ still correlated with $\mathbf{U}$. Therefore, before the outcome regression, we need to estimate the selection score for collider bias correction.

According to Equation (2), estimating the selection score requires estimating $\mathbb{E}[h_2(\mathbf{X}, \hat{T}, \Phi_s) \mid \mathbf{X}, \hat{T}]$. We achieve this estimation by learning a model $f_{h_2}(\mathbf{X}, \hat{T})$ with the following loss function.

$$\mathcal{L}_{f_{h_2}} = \frac{1}{n} \sum_{i=1}^{n} (h_2(\mathbf{x}_i, \hat{t}_i, \hat{f}_{\Phi_s}(\mathbf{d}_i)) - f_{h_2}(\mathbf{x}_i, \hat{t}_i))^2.$$

Afterwards, we estimate the selection score based on Equations (1) and (2) with the following loss function.

$$\mathcal{L}_\pi = \frac{1}{n_1} \sum_{i:s_i=1} \left( g\left(\mathbf{x}_i, \hat{t}_i, y_i\right) \cdot \left( h_2\left(\mathbf{x}_i, \hat{t}_i, \hat{f}_{\Phi_s}(\mathbf{d}_i)\right) - f_{h_2}\left(\mathbf{x}_i, \hat{t}_i\right) \right)^2 \cdot \pi\left(\mathbf{x}_i, \hat{t}_i, y_i, \hat{f}_{\Phi_s}(\mathbf{d}_i)\right)^{-1} \right)^2$$

$$+ \frac{1}{n} \sum_{i=1}^{n} \left( s_i \cdot \pi\left(\mathbf{x}_i, \hat{t}_i, y_i, \hat{f}_{\Phi_s}(\mathbf{d}_i)\right)^{-1} - 1 \right) \cdot h_1\left(\mathbf{x}_i, \hat{t}_i, \hat{f}_{\Phi_s}(\mathbf{d}_i)\right)^2,$$

where $n_1$ denotes the number of samples with $S = 1$ in $\mathbb{D}$, and the basis functions $h_1, h_2$, and $g$ are automatically learned by deep representation learning Xu et al. (2021). Additionally, we adopt the Double/Debiased Machine Learning (DML) framework (Chernozhukov et al., 2018) to mitigate regularization bias and overfitting issues caused by the use of deep learning.

## 4.2 THE SECOND-STAGE OF DDIV

In the second stage of DDIV, we first reweight the samples with $S = 1$ in $\mathcal{D}$ using the learned selection score to recover the full distribution and remove collider bias. In the reweighted samples, which is free from collider bias, $\hat{T}$ is no longer correlated with $\mathbf{U}$. As a result, we can now regress $Y$

Table 1: Out-of-sample MSE (mean ± std) on Demand datasets. Left: varying coupled biases strength $\alpha$ with fixed collider bias strength $\beta = 1.0$. Right: varying collider bias strength $\beta$ with fixed coupled biases strength $\alpha = 1.0$. Results are scaled by a factor of $10^3$, and best results are in bold.

| Method | $\alpha = 0.5$ | $\alpha = 1.0$ | $\alpha = 1.5$ | Method | $\beta = 0.5$ | $\beta = 1.0$ | $\beta = 1.5$ |
|---|---|---|---|---|---|---|---|
| Heckit | 0.553±0.043 | 0.677±0.082 | 0.687±0.092 | Heckit | 0.625±0.058 | 0.677±0.082 | 0.785±0.102 |
| 2SRI | 0.480±0.054 | 0.571±0.075 | 0.653±0.091 | 2SRI | 0.510±0.068 | 0.571±0.075 | 0.614±0.091 |
| IPSW | 0.550±0.058 | 0.630±0.052 | 0.698±0.083 | IPSW | 0.492±0.034 | 0.630±0.052 | 0.723±0.103 |
| DeepIV | 1.062±0.019 | 1.073±0.016 | 1.126±0.041 | DeepIV | 1.054±0.025 | 1.073±0.016 | 1.187±0.034 |
| SIV | 0.483±0.054 | 0.533±0.091 | 0.583±0.263 | SIV | 0.473±0.086 | 0.533±0.091 | 0.790±0.134 |
| KernelIV | 1.007±0.054 | 1.022±0.073 | 1.042±0.098 | KernelIV | 1.011±0.059 | 1.022±0.073 | 1.053±0.109 |
| DeepGMM | 0.982±0.044 | 0.994±0.059 | 1.025±0.087 | DeepGMM | 0.714±0.041 | 0.994±0.059 | 1.123±0.061 |
| DFIV | 1.049±0.051 | 1.071±0.074 | 1.062±0.081 | DFIV | 0.942±0.033 | 1.071±0.074 | 1.094±0.052 |
| CBIV | 0.982±0.024 | 1.011±0.022 | 0.996±0.020 | CBIV | 0.972±0.015 | 1.011±0.022 | 1.052±0.029 |
| 2SSI | 0.361±0.019 | 0.387±0.017 | 0.654±0.019 | 2SSI | 0.569±0.127 | 0.387±0.017 | 0.710±0.423 |
| **DDIV** | **0.192±0.069** | **0.242±0.085** | **0.292±0.126** | **DDIV** | **0.239±0.078** | **0.242±0.085** | **0.384±0.182** |

on $\hat{T}$ using the reweighted samples to address confounding bias based on the IV theory. Specifically, we learn an outcome estimation model $f_y(\mathbf{X}, \hat{T}, \Phi_s)$ with the following loss function.

$$\mathcal{L}_y = \frac{1}{n_1} \sum_{i=1}^{n_1} \frac{(y_i - f_y(\mathbf{x}_i, \hat{t}_i, f_{\Phi_s}(\mathbf{d}_i)))^2}{\pi(\mathbf{x}_i, \hat{t}_i, f_{\Phi_s}(\mathbf{d}_i), y_i)}$$

Based on Theorem 3.4, the learned $\hat{f}_y(\mathbf{X}, \hat{T}, \Phi_s)$ is an unbiased estimate of $\mathbb{E}[Y \mid \mathbf{X}, \hat{T}, \Phi_s]$. Consequently, we can obtain an unbiased estimate of $\mathbb{E}[Y \mid do(T = t), \mathbf{X}]$ based on $\mathbb{P}(\Phi_s)$ and the estimated $\mathbb{E}[Y \mid \mathbf{X}, \hat{T}, \Phi_s]$. The pseudo-code of DDIV is in Algorithm 1 and the source code is provided in the supplementary material. The computational complexity analysis of DDIV is in Appendix H.

## 5 EXPERIMENTS

### 5.1 BASELINES

Since no existing methods can address coupled confounding and collider biases, we compare DDIV with baselines grouped by their target bias. The first group includes IV methods designed to address confounding bias: (1) Two-Stage Residual Inclusion (2SRI) (Terza et al., 2008), (2) DeepIV (Hartford et al., 2017), (3) KernelIV (Singh et al., 2019), (4) DeepGMM (Bennett et al., 2019), (5) DFIV (Xu et al., 2021), and (6) CBIV (Wu et al., 2022). The second group includes methods designed to address collider bias: (1) Heckit (Heckman, 1979), (2) Inverse Probability of Sampling Weights (IPSW) (Cole and Stuart, 2010), and (3) Selection IV (SIV) estimation (Sun et al., 2018). We also compare the proposed method with 2SSI (Li et al., 2024), which is designed to address uncoupled confounding and collider biases. In each experiment, we performed 10 repetitions and recorded the mean and standard deviation (std) of the results. More implementation details are in Appendix D.

### 5.2 EXPERIMENTS ON SYNTHETIC DATA

#### 5.2.1 DATASETS

To evaluate the performance of DDIV, we conducted experiments on a benchmark semi-synthetic dataset, the **Demand** dataset, which is commonly used in IV studies with both low-dimensional and high-dimensional settings (Hartford et al., 2017; Wu et al., 2022; Li et al., 2024). Since the original Demand dataset only contains confounding bias, we followed Li et al. (2024) to introduce collider bias and couple the two biases by generating $S$ based on a function of $\mathbf{U}, \mathbf{X}, T, \mathbf{D}$, and $Y$. We control the effect of $\mathbf{U}$ on $S$ by a hyperparameter $\alpha$ and the effect of $Y$ on $S$ by a hyperparameter $\beta$. The smaller the hyperparameters, the weaker the effects. The detailed data generation process of the coupled-biased Demand dataset is in Appendix E.

### 5.2.2 RESULTS

In our experiments on the Demand dataset, following previous work Hartford et al. (2017); Xu et al. (2021), we use out-of-sample Mean Square Error (MSE) as the evaluation metric. We adjusted $\alpha$, which controls the strength of the coupling between the two biases ($\mathbf{U} \rightarrow S$), and $\beta$, which controls the strength of collider bias ($Y \rightarrow S$), to evaluate the performance under varying bias strengths. Specifically, we first fixed $\beta$ at 1.0 and set $\alpha$ at values of $\{0.5, 1.0, 1.5\}$, and then fixed $\alpha$ at 1.0 and set $\beta$ at values of $\{0.5, 1.0, 1.5\}$.

The experimental results, as shown in Tables 1, lead to the following conclusions: (1) As the strength of the coupled biases ($\alpha$) or collider bias ($\beta$) increases, most models show worse performance, demonstrating the negative impact of such biases on causal estimation. (2) Among all the baselines, methods targeting selection bias (e.g., Heckit, IPSW, SIV) outperform IV-based baselines due to the violation of IV assumptions caused by coupled biases, making the IV invalid and leading to error accumulation from both biases throughout the estimation process. (3) As an ablation of DDIV, SIV effectively addresses collider bias and there outperforms the other baselines. However, it ignores confounding bias, resulting in worse performance than DDIV. (4) 2SSI can only handle non-coupled biases as it assumes $\mathbf{U} \perp\!\!\!\perp S$, which limits its effectiveness in our problem compared to DDIV. (5) DDIV achieves the best results across all settings, which demonstrates its effectiveness under different strengths of biases. The higher variance observed in DDIV is attributed to the inherent instability of the IPW framework. If the estimation of the selection score is inaccurate, IPW can lead to the accumulation of errors across the two stages, thereby affecting the stability of the model.

More experimental results, including experiments on the high-dimensional setting of the Demand dataset and ablation studies, are provided in Appendix F.

### 5.3 EXPERIMENTS ON REAL-WORLD DATA

#### 5.3.1 DATASETS

In this section, we validate the effectiveness of DDIV in practical scenarios using a real-world dataset with well-defined IVs, i.e., the **Wage2 dataset** (Wooldridge, 2016). Following previous work Li et al. (2024), we manually introduce coupled biases through non-random sampling based on certain variables. As there is no ground truth of counterfactual outcomes, our goal becomes estimating the Average Treatment Effect (ATE) for the entire dataset using only the samples with $S = 1$. The detailed description of the coupled-biased Wage2 dataset is in Appendix E.

#### 5.3.2 RESULTS

Following previous work Ding et al. (2017); Li et al. (2024), we use the results from the previous study Wooldridge (2016) as a benchmark result and compare the estimated ATEs with it to evaluate the accuracy of ATE estimation, i.e., methods with results closer to the benchmark result perform better.

The results are reported in Figure 3, with the expected ATE highlighted using a red dashed line. Based on the results, we have the following conclusions: (1) The benchmark ATE interval is positive with a mean of $0.061$. However, many baselines yield estimates close to zero, failing to provide meaningful estimates due to coupled biases. (2) IPSW, 2SRI, DFIV, and 2SSI have obtained ATE values larger than $0.061$, but ex-

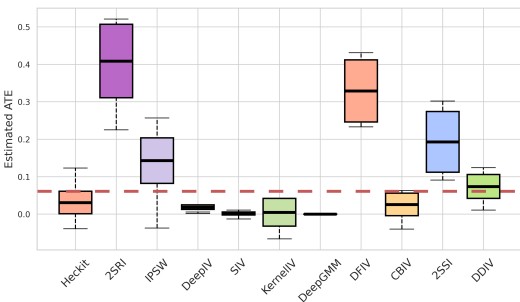

Figure 3: The ATE estimation results on the Wage2 dataset, where the red dashed line represents the ATE reported in Wooldridge (2016).

hibit significant variances, meaning that these methods cannot achieve unbiased ATE estimation under coupled biases. (3) DDIV achieves the best performance, with its mean ATE closest to the benchmark result, which demonstrates that DDIV successfully mitigates the coupled biases.

We also performed sensitivity analyses to demonstrate that the IV set in Wage2 satisfies Assumption 3.1, as reported in Appendix G.

# 6 CONCLUSION

In this paper, we study an unresolved challenge in causal inference, i.e., the coupled confounding and collider biases. To address this challenge, we first propose a novel identification theory for treatment effects under coupled biases with an IV set, which contains subsets satisfying the IV and SIV conditions, respectively. Based on the developed identification theory, we propose a novel method with theoretical guarantees, namely DDIV, which achieves an unbiased estimate of treatment effects under coupled biases. To the best of our knowledge, it is the first work that can address this challenge. Extensive experiments on synthetic and real-world datasets show that DDIV outperforms existing methods under coupled biases, demonstrating its effectiveness.

DDIV has two main limitations: (1) Similar to most IV methods, finding an IV set requires expert knowledge. Whether existing methods for IV testing can be extended to the coupled biases setting is also an interesting problem. (2) Although the correctness of the decomposed representations is theoretically guaranteed, it still requires efforts to choose suitable hyperparameters and model architectures in practice.

## REPRODUCIBILITY STATEMENT

The assumptions and theories underlying this paper are presented in Sections 2 and 3, while the relevant proofs are included in Appendix B. The implementation details of the proposed method are available in Appendix D, and information about the datasets is provided in Appendix E. The source code is uploaded to the supplementary materials.

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

# A  RELATED WORK

**Methods for confounding bias.** Confounding bias is a significant challenge in the field of causal inference, leading to the development of various methods to address this issue. The instrumental variable approach represents a well-established framework. There are many variations of the instrumental variable methods. The most classical is the Two-Stage Least Squares (2SLS), which is based on the linear model assumption. In linear scenarios, the 2SLS method regresses the treatment on the IV in the first stage and then regresses the outcome on the predicted treatment to eliminate the bias effects of unmeasured confounders (Angrist et al., 1996; Angrist and Krueger, 2001; Baiocchi et al., 2014; Brito and Pearl, 2012). Linear scenarios are inadequate for complex real-world situations. Therefore, many approaches are proposed to extend the IV framework to non-linear scenarios. DeepIV addresses confounding bias under non-linear conditions by learning deep representations of variables through deep neural networks (Hartford et al., 2017). KernelIV extends causal relationships to the nonlinear domain by applying kernel techniques in high-dimensional feature spaces (Singh et al., 2019). Recent machine learning methods have also extended the use of instrumental variables to more complex scenarios (Bennett et al., 2019; Xu et al., 2021; Yuan et al., 2022). However, these methods still only focus on addressing confounding bias. In real-world scenarios, it is common for multiple types of biases to exist simultaneously. This paper aims to address the situation where confounding bias and collider bias are coupled. Coupled biases result in the instrumental variables not satisfying the necessary assumptions, thus making it impossible to achieve an unbiased estimation. Other methods focusing on confounding bias, such as negative control and data fusion methods, also cannot deal with coupled biases, as collider bias makes the treatment effect on the full data unidentifiable (Shi et al., 2020; Colnet et al., 2024).

**Methods for selection bias.** Selection bias is also a very common challenge in causal inference, resulting from non-random sample selection. Collider bias is a specific case of selection bias that occurs when both the treatment and the outcome simultaneously influence the sample selection mechanism (Munafò et al., 2018). Observational data in real life is inevitably incomplete and it's hard to guarantee random sampling during data collecting. If there is no selection bias, we can easily estimate the treatment effect using the fully observed data. However, if selection bias exists, without additional assumptions, we cannot estimate the values for the missing outcomes unbiasedly. To address the issue of sample selection bias, the Heckman sample selection model is proposed to estimate a control term to control the impact of selection bias on the outcome estimation (Heckman, 1979; Heckman et al., 2006). The Inverse Probability of Sampling Weights (IPSW) method addresses non-random sample selection by matching the distributions of selected and unselected samples using $\mathbb{P}(S = 1 \mid \mathbf{X})$ (Cole and Stuart, 2010). The above methods assume that the sample selection is caused by only $\mathbf{X}$ and $T$. As a result, they cannot address collider bias due to the presence of $Y \to S$. Recently, some studies have proposed using shadow variables to address collider bias by leveraging the odds ratio function to estimate the distribution of the outcome conditional on $S = 0$ (d'Haultfoeuille, 2010; Miao and Tchetgen Tchetgen, 2016; Li et al., 2023; Miao et al., 2024). As proposed in previous work Sun et al. (2018), the authors propose a Selection IV (SIV) method that addresses collider bias by identifying $\mathbb{P}(\mathbf{X}, T, Y)$ for doubly robust estimation. However, these methods assume that all the confounders are observable. Once some confounding factors $\mathbf{U}$ are unobserved, all the methods become ineffective in unbiased treatment effect estimation. Recently, Li et al. (2024) have proposed the identification theory and estimation method for treatment effects in the presence of both unobserved confounders and collider bias. However, they assume that unmeasured confounders do not directly affect $S$ ($\mathbf{U} \not\to S$), implying that the two biases are not coupled. Therefore, their theory and method cannot apply to coupled biases. To the best of our knowledge, there are currently no methods that can address the problem of coupled biases.

# B  PROOFS

## B.1  PROOF OF THEOREM 2.8

The proof is based on contradiction.

Based on Definition 2.5, $V \perp\!\!\!\perp Y \mid \mathbf{X}, T$, we have

$$\mathbb{P}(\mathbf{X}, T, Y, V, S; \theta_i, \xi_i, \upsilon_i) = \mathbb{P}(S \mid \mathbf{X}, T, Y, V; \theta_i) \cdot \mathbb{P}_{\upsilon_i}(V \mid \mathbf{X}, T; \upsilon_i) \cdot \mathbb{P}(\mathbf{X}, T, Y; \xi_i),$$

where $\theta_i$, $\upsilon_i$, and $\xi_i$ denote candidate parameters for $\mathbb{P}(\mathbf{X}, T, Y, V, S)$.

If $\mathbb{P}(\mathbf{X}, T, Y, V, S)$ cannot be identified, $\exists \theta_1 \neq \theta_2$, $\upsilon_1 \neq \upsilon_2$, and $\xi_1 \neq \xi_2$ satisfying that

$$\mathbb{P}(S \mid \mathbf{X}, T, Y, V; \theta_1) \cdot \mathbb{P}(V \mid \mathbf{X}, T; \upsilon_1) \cdot \mathbb{P}(\mathbf{X}, T, Y; \xi_1)$$
$$= \mathbb{P}(S \mid \mathbf{X}, T, Y, V; \theta_2) \cdot \mathbb{P}(V \mid \mathbf{X}, T; \upsilon_2) \cdot \mathbb{P}(\mathbf{X}, T, Y; \xi_2).$$

This equation is equivalent to

$$\frac{\mathbb{P}(S \mid \mathbf{X}, T, Y, V; \theta_1)}{\mathbb{P}(S \mid \mathbf{X}, T, Y, V; \theta_2)} = \frac{\mathbb{P}(V \mid \mathbf{X}, T; \upsilon_2) \cdot \mathbb{P}(\mathbf{X}, T, Y; \xi_2)}{\mathbb{P}(V \mid \mathbf{X}, T; \upsilon_1) \cdot \mathbb{P}(\mathbf{X}, T, Y; \xi_1)},$$

Suppose $\upsilon_1$, $\upsilon_2$, $\xi_1$, and $\xi_2$ satisfy that

$$\mathbb{P}(V \mid \mathbf{X}, T; \upsilon_1) = \mathbb{P}(V \mid \mathbf{X}, T; \upsilon_2),$$
$$\mathbb{P}(\mathbf{X}, T, Y; \xi_1) \neq \mathbb{P}(\mathbf{X}, T, Y; \xi_2).$$

Then, we have

$$\frac{\mathbb{P}(S \mid \mathbf{X}, T, Y, V; \theta_1)}{\mathbb{P}(S \mid \mathbf{X}, T, Y, V; \theta_2)} = \frac{\mathbb{P}(\mathbf{X}, T, Y; \xi_2)}{\mathbb{P}(\mathbf{X}, T, Y; \xi_1)},$$

which varies with $\mathbf{X}$, $T$, and $Y$.

This contradicts Condition 2.7, which requires that $\forall \theta_1 \neq \theta_2$, the ratio $\frac{\mathbb{P}(S \mid \mathbf{X}, T, Y, V; \theta_1)}{\mathbb{P}(S \mid \mathbf{X}, T, Y, V; \theta_2)}$ is either a constant or varies with $\mathbf{X}$, $T$, and $V$.

## B.2  PROOF OF THEOREM 3.3

To prove that $\mathbb{E}[Y \mid do(T = t), \mathbf{X}]$ is identifiable, we first prove that $\mathbb{P}(\mathbf{X}, T, Y, \mathbf{D}_s)$ is identifiable, then demonstrate that $\mathbb{E}[Y \mid do(T = t), \mathbf{X}, \mathbf{D}_s]$ can be identified from $\mathbb{P}(\mathbf{X}, T, Y, \mathbf{D}_s)$ with the help of $\mathbf{D}_t$, and finally, $\mathbb{E}[Y \mid do(T = t), \mathbf{X}]$ is identified based on $\mathbb{E}[Y \mid do(T = t), \mathbf{X}, \mathbf{D}_s]$ and $\mathbb{P}(\mathbf{D}_s)$.

Based on Theorem 2.8, a sufficient condition for the identification of $\mathbb{P}(\mathbf{X}, T, Y, \mathbf{D}_s)$ is Condition 2.7. Therefore, we begin by proving that, under Assumptions 2.1, 2.2, 3.1, and 3.2, Condition 2.7 is satisfied, thus ensuring the identification of $\mathbb{P}(\mathbf{X}, T, Y, \mathbf{D}_s)$. The detailed proof for the satisfaction of Condition 2.7 is as below. The proof is based on contradiction.

Without loss of generality, we let $V$ denote $\mathbf{D}_s$, omit $\mathbf{X}$ and $T$ from the functions $h$ and $q$ in Assumption 3.2, as they represent common conditional sets in $h$ and $q$, and assume that $V$, $Y$, and $U$ are one-dimensional. Based on Assumption 3.1, we have $V \perp\!\!\!\perp U, Y$. Consider the case where $V$, $Y$, and $U$ are continuous random variables.

Under Assumptions 2.1, 2.2, and 3.2, if Condition 2.7 does not hold and $\mathbb{P}(V, Y, U)$ cannot be identified, $\exists \zeta_1 \neq \zeta_2$, and $\omega_1 \neq \omega_2$, and $\xi_1 \neq \xi_2$ satisfying that $\mathbb{P}(Y, U; \xi_1) = \mathbb{P}(Y, U; \xi_2)$ and $\frac{\pi(Y, V; \zeta_1, \omega_1)}{\pi(Y, V; \zeta_2, \omega_2)}$ varies with $U$ and $Y$, i.e.,

$$\frac{\text{expit}(q(V; \zeta_1) + h(Y, U; \omega_1))}{\text{expit}(q(V; \zeta_2) + h(Y, U; \omega_2))} = g(Y, U),$$

where $\zeta_1$ and $\zeta_2$ denote two sets of different candidate parameters for $q(V)$, $\zeta_1$ and $\zeta_2$ denote two sets of different candidate parameters for $h(Y, U)$, $\text{expit}(\cdot)$ denotes the logistic function, and $g(Y, U) = \mathbb{P}(Y, U; \xi_2)/\mathbb{P}(Y, U; \xi_1)$.

Based on Assumption 3.1, taking derivatives with respect to $V$ on both sides gives

$$\frac{\frac{\partial}{\partial V}\text{expit}(q(V;\zeta_1) + h(Y,U;\omega_1))}{\text{expit}(q(V;\zeta_1) + h(Y,U;\omega_1))} = \frac{\frac{\partial}{\partial V}\text{expit}(q(V;\zeta_2) + h(Y,U;\omega_2))}{\text{expit}(q(V;\zeta_2) + h(Y,U;\omega_2))},$$

which is equivalent to

$$\frac{\partial q(V;\zeta_1)}{\partial V}(1 + \exp(q(V;\zeta_2) + h(Y,U;\omega_2))) = \frac{\partial q(V;\zeta_2)}{\partial V}(1 + \exp(q(V;\zeta_1) + h(Y,U;\omega_1))), \quad (3)$$

where $\exp(\cdot)$ denotes the exponential function.

Taking derivatives with respect to $Y$ on both sides leads to

$$\frac{\partial q(V;\zeta_1)/\partial V}{\partial q(V;\zeta_2)/\partial V}\exp(q(V;\zeta_2) - q(V;\zeta_1)) = \frac{\partial h(Y,U;\omega_1)/\partial Y}{\partial h(Y,U;\omega_2)/\partial Y}\exp(h(Y,U;\omega_1) - h(Y,U;\omega_2))).$$

The left-hand side depends only on $V$, and the right-hand side depends only on $Y$ and $U$. Since $V \perp\!\!\!\perp U, Y$, it must be that

$$\frac{\partial q(V;\zeta_1)/\partial V}{\partial q(V;\zeta_2)/\partial V}\exp(q(V;\zeta_2)-q(V;\zeta_1)) = \frac{\partial h(Y,U;\omega_1)/\partial Y}{\partial h(Y,U;\omega_2)/\partial Y}\exp(h(Y,U;\omega_1)-h(Y,U;\omega_2))) = c_1,$$

where $c_1$ is a constant. Equivalently, we have

$$\frac{\partial q(V;\zeta_1)/\partial V}{\partial q(V;\zeta_2)/\partial V} = \frac{c_1}{\exp(q(V;\zeta_2) - q(V;\zeta_1))}.$$

Substituting the above equation into Equation (3), we have

$$\frac{c_1 \cdot (1 + \exp(q(V;\zeta_2) + h(Y,U;\omega_2)))}{\exp(q(V;\zeta_2) - q(V;\zeta_1))} = 1 + \exp(q(V;\zeta_1) + h(Y,U;\omega_1)),$$

which is equivalent to

$$c_1 \cdot (\exp(-q(V;\zeta_2)) + \exp(h(Y,U;\omega_2))) = \exp(-q(V;\zeta_1)) + \exp(h(Y,U;\omega_1)).$$

Therefore, $\exists c_2$ satisfying that

$$c_1 \cdot \exp(-q(V;\zeta_2)) + c_2 = \exp(-q(V;\zeta_1)) \quad (4)$$

and

$$c_1 \cdot \exp(h(Y,U;\omega_2)) - c_2 = \exp(h(Y,U;\omega_1)), \quad (5)$$

where $c_2$ is a constant. From Equations (4) and (5), we have

$$c_1 = \frac{\exp(-q(V;\zeta_1)) + \exp(h(Y,U;\omega_1))}{\exp(-q(V;\zeta_2)) + \exp(h(Y,U;\omega_2))}, \quad (6)$$

Substituting Equations (4), (5), and (6) into $\frac{\pi(Y,V;\zeta_1,\omega_1)}{\pi(Y,V;\zeta_2,\omega_2)}$, we have

$$\frac{\text{expit}(q(V;\zeta_1) + h(Y,U;\omega_1))}{\text{expit}(q(V;\zeta_2) + h(Y,U;\omega_2))}$$

$$= \frac{\exp(q(V;\zeta_1) + h(Y,U;\omega_1)) \cdot (1 + \exp(q(V;\zeta_2) + h(Y,U;\omega_2)))}{\exp(q(V;\zeta_2) + h(Y,U;\omega_2)) \cdot (1 + \exp(q(V;\zeta_1) + h(Y,U;\omega_1)))}$$

$$= \frac{\exp(q(V;\zeta_1) + h(Y,U;\omega_1)) + \exp(q(V;\zeta_1) + h(Y,U;\omega_1) + q(V;\zeta_2) + h(Y,U;\omega_2))}{\exp(q(V;\zeta_2) + h(Y,U;\omega_2)) + \exp(q(V;\zeta_2) + h(Y,U;\omega_2) + q(V;\zeta_1) + h(Y,U;\omega_1))}$$

$$= \frac{\exp(-q(V;\zeta_2) + h(Y,U;\omega_1)) + \exp(h(Y,U;\omega_1) + h(Y,U;\omega_2)))}{\exp(-q(V;\zeta_1) + h(Y,U;\omega_2)) + \exp(h(Y,U;\omega_2) + h(Y,U;\omega_1)))}$$

$$= \frac{\exp(h(Y,U;\omega_1))}{\exp(h(Y,U;\omega_2))} \cdot \frac{\exp(-q(V;\zeta_2)) + \exp(h(Y,U;\omega_2))}{\exp(-q(V;\zeta_1)) + \exp(h(Y,U;\omega_1))}$$

$$= \frac{\exp(h(Y,U;\omega_1))}{\exp(h(Y,U;\omega_2)) \cdot c_1}$$

$$= \frac{\exp(h(Y,U;\omega_1))}{c_2 + \exp(h(Y,U;\omega_1))}$$

$$= (1 + c_2 \cdot \exp(-h(Y,U;\omega_1)))^{-1}$$

$$= g(Y,U).$$

Since $g(Y, U) = \mathbb{P}(Y, U; \xi_2)/\mathbb{P}(Y, U; \xi_1)$ and $\mathbb{P}(Y, U; \xi_1) = \mathbb{P}(Y, U; \xi_2)$, it must be that $c_2 = 0$, leading to a contradiction with the variation of $\frac{\pi(Y, V; \zeta_1, \omega_1)}{\pi(Y, V; \zeta_2, \omega_2)}$ with respect to $Y$ and $U$. Therefore, $\frac{\pi(Y, V; \zeta_1, \omega_1)}{\pi(Y, V; \zeta_2, \omega_2)}$ is either a constant or varies with $V$, proving that Condition 2.7 holds.

Consequently, since the sufficient condition for the identification of $\mathbb{P}(\mathbf{X}, T, Y, \mathbf{D}_s)$ holds under Assumptions 2.1, 2.2, 3.1, and 3.2, $\mathbb{P}(\mathbf{X}, T, Y, \mathbf{D}_s)$ is proven to be identifiable. Then, under Assumptions 2.1, 2.2, 3.1, and 2.4, based on the IV theory (Angrist et al., 1996; Hernán and Robins, 2006; Hartwig et al., 2023), $\mathbb{E}[Y \mid do(T = t), \mathbf{X}, \mathbf{D}_s]$ can be identified from $\mathbb{P}(\mathbf{X}, T, Y, \mathbf{D}_s)$ with the help of $\mathbf{D}_t$.

Finally, based on Assumption 3.1, $\mathbf{D}_s \perp\!\!\!\perp \mathbf{U}, \mathbf{X}, T$. Therefore, $\mathbb{E}[Y \mid do(T = t), \mathbf{X}]$ is identified as

$$\mathbb{E}[Y \mid do(T = t), \mathbf{X}] = \int \mathbb{E}[Y \mid do(T = t), \mathbf{X}, \mathbf{D}_s] \, d\mathbb{P}(\mathbf{D}_s \mid do(T = t), \mathbf{X})$$

$$= \int \mathbb{E}[Y \mid do(T = t), \mathbf{X}, \mathbf{D}_s] \, d\mathbb{P}(\mathbf{D}_s)$$

### B.3 Proof of Theorem 3.4

We prove the unbiasedness of the selection score estimator and the outcome estimator, and then the consistency and asymptotic normality of the two estimators in large samples holds under the conditions specified in previous work (Newey and McFadden, 1994; Newey and Powell, 2003; Sun et al., 2018).

**Unbiasedness of the selection score estimator.** First, let $\upsilon^*$ denote the true parameters of $\mathbb{E}[h_2(\mathbf{X}, T, \mathbf{D}_s) \mid \mathbf{X}, T]$, if $\mathbb{E}[h_2(\mathbf{X}, T, \mathbf{D}_s) \mid \mathbf{X}, T; \upsilon]$ is correctly specified, we have

$$\mathbb{E}[h_2(\mathbf{X}, T, \mathbf{D}_s) - \mathbb{E}[h_2(\mathbf{X}, T, \mathbf{D}_s) \mid \mathbf{X}, T; \upsilon^*]]$$
$$= \mathbb{E}[\mathbb{E}[h_2(\mathbf{X}, T, \mathbf{D}_s) - \mathbb{E}[h_2(\mathbf{X}, T, \mathbf{D}_s) \mid \mathbf{X}, T; \upsilon^*]|\mathbf{X}, T]]$$
$$= \mathbb{E}[\mathbb{E}[h_2(\mathbf{X}, T, \mathbf{D}_s) \mid \mathbf{X}, T; \upsilon^*] - \mathbb{E}[h_2(\mathbf{X}, T, \mathbf{D}_s) \mid \mathbf{X}, T; \upsilon^*]]$$
$$= 0.$$

Therefore, $\mathbb{E}[h_2(\mathbf{X}, T, \mathbf{D}_s) \mid \mathbf{X}, T; \upsilon]$ is unbiased.

Next, let $\zeta^*$ and $\omega^*$ denote the true parameters of $\pi(\mathbf{X}, T, Y, \mathbf{D}_s)$, if $\pi(\mathbf{X}, T, Y, \mathbf{D}_s; \zeta, \omega)$ is correctly specified, we have

$$\mathbb{E}[\psi(\mathbf{X}, T, \mathbf{D}_s; \zeta^*, \omega^*)] = \mathbb{E}\left[\left(\frac{S}{\pi(\mathbf{X}, T, Y, \mathbf{D}_s; \zeta^*, \omega^*)} - 1\right) \cdot h_1(\mathbf{X}, T, \mathbf{D}_s)\right]$$

$$= \mathbb{E}\left[\mathbb{E}\left[\left(\frac{S}{\pi(\mathbf{X}, T, Y, \mathbf{D}_s; \zeta^*, \omega^*)} - 1\right) \cdot h_1(\mathbf{X}, T, \mathbf{D}_s)\Big|\mathbf{X}, T, Y, \mathbf{D}_s\right]\right]$$

$$= \mathbb{E}\left[\left(\frac{\pi(\mathbf{X}, T, Y, \mathbf{D}_s; \zeta^*, \omega^*)}{\pi(\mathbf{X}, T, Y, \mathbf{D}_s; \zeta^*, \omega^*)} - 1\right) \cdot h_1(\mathbf{X}, T, \mathbf{D}_s)\right]$$

$$= 0.$$

Moreover, since $\mathbf{D}_s \perp\!\!\!\perp Y \mid \mathbf{X}, T$ is satisfied, we have

$$\mathbb{E}[\delta(\mathbf{X}, T, Y, \mathbf{D}_s; \zeta^*, \omega^*)]$$
$$= \mathbb{E}\left[(h_2(\mathbf{X}, T, \mathbf{D}_s) - \mathbb{E}[h_2(\mathbf{X}, T, \mathbf{D}_s) \mid \mathbf{X}, T; \upsilon^*]) \cdot \frac{S \cdot g(\mathbf{X}, T, Y)}{\pi(\mathbf{X}, T, Y, \mathbf{D}_s; \zeta^*, \omega^*)}\right]$$
$$= \mathbb{E}[(h_2(\mathbf{X}, T, \mathbf{D}_s) - \mathbb{E}[h_2(\mathbf{X}, T, \mathbf{D}_s) \mid \mathbf{X}, T; \upsilon^*]) \cdot g(\mathbf{X}, T, Y)]$$
$$= \mathbb{E}[\mathbb{E}[(h_2(\mathbf{X}, T, \mathbf{D}_s) - \mathbb{E}[h_2(\mathbf{X}, T, \mathbf{D}_s) \mid \mathbf{X}, T; \upsilon^*]) \cdot g(\mathbf{X}, T, Y)|\mathbf{X}, T]]$$
$$= \mathbb{E}[(\mathbb{E}[h_2(\mathbf{X}, T, \mathbf{D}_s)|\mathbf{X}, T; \upsilon^*] - \mathbb{E}[h_2(\mathbf{X}, T, \mathbf{D}_s) \mid \mathbf{X}, T; \upsilon^*]) \cdot \mathbb{E}[g(\mathbf{X}, T, Y) \mid \mathbf{X}, T]]$$
$$= 0.$$

Therefore, $\pi(\mathbf{X}, T, Y, \mathbf{D}_s; \zeta, \omega)$ is unbiased.

**Unbiasedness of the outcome estimator.** The theory of unbiasedness for the two-stage IV regression is well-established (Newey and Powell, 2003; Heckman et al., 2006; Hernán and Robins, 2006;

Hartwig et al., 2023), i,e.,

$$\mathbb{E}\left[Y - \mathbb{E}\left[Y \mid \mathbf{X}, \hat{T}, \mathbf{D}_s; \xi^*\right]\right] = 0,$$

where $\xi^*$ denotes the true parameters of $\mathbb{E}[Y \mid \mathbf{X}, \hat{T}, \mathbf{D}_s]$.

Here, we use the above conclusion to prove the unbiasedness of the IV-based IPW estimator. If $\mathbb{E}[Y \mid \mathbf{X}, \hat{T}, \mathbf{D}_s; \xi]$ is correctly specified, we have

$$\mathbb{E}\left[\frac{S \cdot Y}{\pi(\mathbf{X}, \hat{T}, Y, \mathbf{D}_s; \zeta^*, \omega^*)} - \mathbb{E}\left[Y \mid \mathbf{X}, \hat{T}, \mathbf{D}_s; \xi^*\right]\right]$$

$$= \mathbb{E}\left[\mathbb{E}\left[\frac{S \cdot Y}{\pi(\mathbf{X}, \hat{T}, Y, \mathbf{D}_s; \zeta^*, \omega^*)} - \mathbb{E}\left[Y \mid \mathbf{X}, \hat{T}, \mathbf{D}_s; \xi^*\right]\bigg| \mathbf{X}, \hat{T}, Y, \mathbf{D}_s\right]\right]$$

$$= \mathbb{E}\left[\frac{\pi\left(\mathbf{X}, \hat{T}, Y, \mathbf{D}_s; \zeta^*, \omega^*\right) \cdot Y}{\pi\left(\mathbf{X}, \hat{T}, Y, \mathbf{D}_s; \zeta^*, \omega^*\right)} - \mathbb{E}\left[Y \mid \mathbf{X}, \hat{T}, \mathbf{D}_s; \xi^*\right]\right]$$

$$= \mathbb{E}\left[Y - \mathbb{E}\left[Y \mid \mathbf{X}, \hat{T}, \mathbf{D}_s; \xi^*\right]\right]$$

$$= 0.$$

Therefore, $\mathbb{E}[Y \mid \mathbf{X}, \hat{T}, \mathbf{D}_s; \xi]$ is unbiased.

Under the conditions specified in previous work (Newey and McFadden, 1994; Newey and Powell, 2003; Sun et al., 2018), $\upsilon^*, \eta^*, \omega^*$, and $\xi^*$ are the probability limits of $\hat{\upsilon}, \hat{\eta}, \hat{\omega}$, and $\hat{\xi}$, respectively. Therefore, the consistency of the two estimators in large samples holds.

### B.4 PROOF OF PROPOSTION 4.1

Under Assumption 3.1, all variables in the IV set satisfy the exclusion and unconfounded conditions for IV. That is, we have $\mathbf{D} \perp\!\!\!\perp Y \mid \mathbf{U}, \mathbf{X}, T$ and $\mathbf{D} \perp\!\!\!\perp \mathbf{U}, \mathbf{X}$. Since $f_{\Phi_t}(\mathbf{D})$ and $f_{\Phi_s}(\mathbf{D})$ are functions of $\mathbf{D}$, the above independence conditions are preserved, i.e., we have $f_{\Phi_t}(\mathbf{D}) \perp\!\!\!\perp Y \mid \mathbf{U}, \mathbf{X}, T$ and $f_{\Phi_t}(\mathbf{D}) \perp\!\!\!\perp \mathbf{U}, \mathbf{X}$, as well as $f_{\Phi_s}(\mathbf{D}) \perp\!\!\!\perp Y \mid \mathbf{U}, \mathbf{X}, T$ and $f_{\Phi_s}(\mathbf{D}) \perp\!\!\!\perp \mathbf{U}, \mathbf{X}$.

Therefore, what remains to be proven is that $f_{\Phi_t}(\mathbf{D})$ also satisfies the relevance condition for IV, and $f_{\Phi_s}(\mathbf{D})$ satisfies the two conditions for SIV. That is, we need to prove that there exist solutions to the objective function of DDIV that satisfy the following conditions: (1) $\hat{f}_{\Phi_t}(\mathbf{D}) \not\perp\!\!\!\perp T \mid \mathbf{X}$, (2) $\hat{f}_{\Phi_s}(\mathbf{D}) \not\perp\!\!\!\perp S \mid \mathbf{X}, T$, and (3) $\hat{f}_{\Phi_s}(\mathbf{D}) \perp\!\!\!\perp Y \mid \mathbf{X}, T$.

If $\lambda_1, \lambda_2$ are sufficiently large, then the objective function of DDIV is as follows.

$$\hat{f}_t, \hat{f}_s, \hat{f}_{\Phi_t}, \hat{f}_{\Phi_s} = \underset{f_t, f_s, f_{\Phi_t}, f_{\Phi_s}}{\arg\min} \; \mathbb{E}[(t_i - f_t(\mathbf{x}_i, f_{\Phi_t}(\mathbf{d}_i)))^2] - \mathbb{E}\left[\log(f_s(\mathbf{x}_i, t_i, f_{\Phi_s}(\mathbf{d}_i)))\right]$$

$$\text{subject to } \mathcal{I}(T, f_{\Phi_s}(\mathbf{D})) = 0 \text{ and } \mathcal{I}(f_{\Phi_s}(\mathbf{D}), f_{\Phi_t}(\mathbf{D})) = 0,$$

If the MI estimator can approximate the upper bound of mutual information, then any solutions to the above objective function must satisfy $\mathcal{I}(T, f_{\Phi_s}(\mathbf{D})) = 0$ and $\mathcal{I}(f_{\Phi_s}(\mathbf{D}), f_{\Phi_t}(\mathbf{D})) = 0$, meaning that $f_{\Phi_s}(\mathbf{D}) \perp\!\!\!\perp T$ and $f_{\Phi_s}(\mathbf{D}) \perp\!\!\!\perp f_{\Phi_t}(\mathbf{D})$.

Therefore, there exist solutions $\hat{f}_{\Phi_t}$ and $\hat{f}_{\Phi_s}$, which satisfy

$$\hat{f}_{\Phi_s} \in \underset{f_{\Phi_s}:\mathcal{I}(T, f_{\Phi_s}(\mathbf{D}))=0}{\arg\max} \; \mathbb{E}\left[\log(f_s(\mathbf{x}_i, t_i, f_{\Phi_s}(\mathbf{d}_i)))\right]$$

and

$$\hat{f}_{\Phi_t} \in \underset{f_{\Phi_t}:\mathcal{I}(f_{\Phi_s}(\mathbf{D}), f_{\Phi_t}(\mathbf{D}))=0}{\arg\min} \; \mathbb{E}[(t_i - f_t(\mathbf{x}_i, f_{\Phi_t}(\mathbf{d}_i)))^2].$$

As a result, if the parameters of the models are sufficiently rich (Hornik et al., 1989), the solutions $\hat{f}_{\Phi_t}$ and $\hat{f}_{\Phi_s}$ satisfy (1) $\hat{f}_{\Phi_t}(\mathbf{D}) \not\perp\!\!\!\perp T \mid \mathbf{X}$ and (2) $\hat{f}_{\Phi_s}(\mathbf{D}) \not\perp\!\!\!\perp S \mid \mathbf{X}, T$.

Finally, since $\hat{f}_{\Phi_s}(\mathbf{D}) \perp\!\!\!\perp \mathbf{U}, \mathbf{X}, T, \Phi_t$ and $\hat{f}_{\Phi_s}(\mathbf{D}) \perp\!\!\!\perp Y \mid \mathbf{U}, \mathbf{X}, T$, we have

$$
\begin{aligned}
\mathbb{P}(\hat{f}_{\Phi_s}(\mathbf{D}), Y \mid \mathbf{X}, T) &= \int \mathbb{P}(\hat{f}_{\Phi_s}(\mathbf{D}), Y, \mathbf{U} \mid \mathbf{X}, T) \, d\mathbf{U} \\
&= \int \mathbb{P}(\hat{f}_{\Phi_s}(\mathbf{D}), Y \mid \mathbf{U}, \mathbf{X}, T) \cdot \mathbb{P}(\mathbf{U} \mid \mathbf{X}, T) \, d\mathbf{U} \\
&= \int \mathbb{P}(\hat{f}_{\Phi_s}(\mathbf{D}) \mid \mathbf{U}, \mathbf{X}, T) \cdot \mathbb{P}(Y \mid \mathbf{U}, \mathbf{X}, T) \cdot \mathbb{P}(\mathbf{U} \mid \mathbf{X}, T) \, d\mathbf{U} \\
&= \mathbb{P}(\hat{f}_{\Phi_s}(\mathbf{D})) \cdot \int \mathbb{P}(Y \mid \mathbf{U}, \mathbf{X}, T) \cdot \mathbb{P}(\mathbf{U} \mid \mathbf{X}, T) \, d\mathbf{U} \\
&= \mathbb{P}(\hat{f}_{\Phi_s}(\mathbf{D})) \cdot \mathbb{P}(Y \mid \mathbf{X}, T) \\
&= \mathbb{P}(\hat{f}_{\Phi_s}(\mathbf{D}) \mid \mathbf{X}, T) \cdot \mathbb{P}(Y \mid \mathbf{X}, T).
\end{aligned}
$$

Therefore, the solution $\hat{f}_{\Phi_s}$ also satisfies (3) $\hat{f}_{\Phi_s}(\mathbf{D}) \perp\!\!\!\perp Y \mid \mathbf{X}, T$.

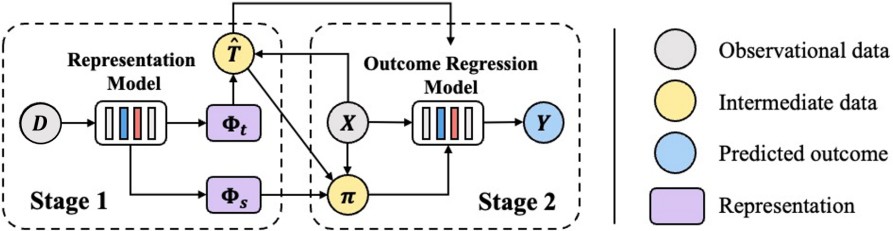

Figure 4: An overview of DualDebiasIV (DDIV). In the first stage, DDIV decomposes the representations satisfying the IV and SIV conditions from the IV sets, respectively. Simultaneously, it performs regression on the treatment using the IV representation and estimates the selection score using the SIV representation. In the second stage, DDIV reweights the sample using the inverse of the selection score and regresses the outcome on the SIV representation, the estimated treatment, and the observed covariates, yielding unbiased treatment effect estimation.

## C PSEUDO-CODE

As stated in Section 3 and illutrated in Figure 4, we propose a novel DDIV method to address the problem of causal inference under coupled confounding and collider biases. The detailed pseudocode is in Algorithm 1.

---

**Algorithm 1** DualDebiasIV

---

**Require:** $\mathbb{D} = \{\mathbf{x}_i, t_i, y_i, \mathbf{d}_i, s_i\}_{i=1}^n$, $\lambda_1, \lambda_2$, mini-batch sizes $m_1, m_2$, number of updates $e_1, e_2$.
**Ensure:** The estimated conditional expectation of $Y$ on the target population, i.e., $\mathbb{E}[Y \mid \mathbf{X}, \hat{T}, \Phi_s]$.
  Initialize parameters in $f_{\Phi_s}, f_{\Phi_t}, f_s, f_t, f_{h_2}, f_p, f_y$.
  **repeat**
    Sample $m_1$ units from $\mathbb{D}$ as Batch 1.
    **for** $j = 1$ to $e_1$ **do**
      Optimize $f_{\Phi_s}, f_{\Phi_t}$ and $f_s$ by $\mathcal{L}_{\Phi}$ using Batch 1.
      Optimize $f_t$ by $\mathcal{L}_t$ using Batch 1.
    **end for**
    $\hat{T} \leftarrow f_t(\mathbf{X}, \Phi_t)$.
    Sample $m_2$ units with $S = 1$ from $\mathbb{D}$ as Batch 2.
    **for** $j = 1$ to $e_2$ **do**
      Optimize $f_{h_2}$ by $\mathcal{L}_{h_2}$ using Batch 1.
      Optimize $f_p$ by $\mathcal{L}_p$ using Batch 2.
    **end for**
    $\pi(\mathbf{X}, \hat{T}, \Phi_s, Y) \leftarrow f_p(\mathbf{X}, \hat{T}, \Phi_s, Y)$
    **for** $j = 1$ to $e_2$ **do**
      Optimize $f_y$ by $\mathcal{L}_y$ using Batch 2.
    **end for**
  **until convergence**
  $\mathbb{E}[Y \mid \mathbf{X}, \hat{T}, \Phi_s] \leftarrow f_y(\mathbf{X}, \hat{T}, \Phi_s)$.
  **return** $\mathbb{E}[Y \mid \mathbf{X}, \hat{T}, \Phi_s]$

---

## D IMPLEMENTATION DETAILS

The experiments were conducted on a Linux Ubuntu 20.04 system with Python 3.8 and PyTorch 1.13.0, utilizing CUDA 12.1 for GPU acceleration. The hardware setup included an AMD EPYC 7763 64-Core Processor for CPU computations and an NVIDIA GeForce RTX 4090 for GPU-based deep learning tasks. This environment provided robust computational capabilities to support the training and evaluation of our models. The implementation details of our algorithm are as follows: Our code framework is based on Xu et al. (2021), leveraging multilayer perceptrons (MLPs) to design the main

Table 2: Hyperparameters of DDIV on different datasets, where Demand$_{\text{Low}}$ denotes Demand datasets under the low-dimensional setting and Demand$_{\text{High}}$ denotes Demand datasets under the high-dimensional setting.

| Setting | Demand$_{\text{Low}}$ | Demand$_{\text{High}}$ | Wage2 |
|---|---|---|---|
| Epoch | 100 | 100 | 100 |
| Learning rate | 0.001 | 0.001 | 0.001 |
| Weight decay | 0.001 | 0.01 | 0.0001 |
| $\lambda_1$ | 0.1 | 0.5 | 0.1 |
| $\lambda_2$ | 0.1 | 0.5 | 0.1 |

architecture of DDIV. The training process utilizes the Adam optimizer (Kingma and Ba, 2015) along with batch normalization (Ioffe and Szegedy, 2015). Following Wu et al. (2022), in the decomposed representation module, we employ Integral Probability Metric (IPM) and Contrastive Log-ratio Upper Bound (CLUB) (Cheng et al., 2020) as mutual information (MI) constraints to enforce independence. In the loss function for decomposed representation, the value of hyperparameters, e.g., $\lambda_1$ and $\lambda_2$, are listed in Table 2. In the formulas, $\text{expit}(x) = \frac{1}{1+e^{-x}}$ denotes the logistic (sigmoid) function, and $\text{logit}(p) = \log\left(\frac{p}{1-p}\right)$ denotes its inverse.

# E    DETAILED DESCRIPTION OF THE DATASETS

## E.1    DEMAND DATASET

In the Demand dataset, the target is to estimate the impact of ticket prices ($T$) on ticket sales ($Y$). It is assumed that there are seven customer types $X \in \{1, \cdots, 7\}$, representing varying sensitivities to price. The parameter $t \sim \text{unif}(0, 10)$ in $\psi_t$ represents time, which is an observable variable included in the dataset. A complex nonlinear function with parameters $t$ and $\psi_t$, is used to model the effect of holidays on ticket prices. The detailed formulation of $\psi_t$ is presented later. Ticket prices ($T$) are modeled as a function of $\psi_t$, fuel prices as well as various randomly distributed discounts ($\mathbf{D}$). Airlines adjust ticket prices dynamically based on fuel prices and discounts as well as holidays to cater to different customer sensitivity groups (Bang and Davis, 2007). We simulated $\mathbf{D} \in \{\mathbf{D}_t, \mathbf{D}_s, \mathbf{D}_{ts}\}$, which represents the IV, SIV, and IV that also influences $S$, respectively. In the experiments conducted on the Demand dataset, unless otherwise specified, the dimension of $\mathbf{D}_t$ is 3, the dimension of $\mathbf{D}_s$ is 5, and the dimension of $\mathbf{D}_{ts}$ is 2. High demand during international conferences introduces confounding bias into the data, represented as an unobservable noise term $V$ in $T$. Additionally, the latent factor $U$ sampled based on a Gaussian noise term $V$ also influences ticket sales ($Y$). The Demand dataset also has a high-dimensional version, where customer sensitivity is replaced with high-dimensional pixel data from the MNIST (LeCun and Cortes, 2010) dataset. We follow the same data generation process as in prior studies (Hartford et al., 2017; Xu et al., 2021).

The original Demand dataset does not inherently exhibit coupled confounding and collider biases. To introduce collider bias, we construct a selection rule to simulate whether the outcome value of a sample is observable (Li et al., 2024). Furthermore, we incorporate the influence of $U$ on $S$ to simulate coupled biases. We define two parameters, $\alpha$ and $\beta$, to control the strength of $U \to S$ and $Y \to S$, respectively. The data generation process is as follows:

$$Y = 100 + (10 + T)X\psi_t - 2T + U,$$

$$T = 25 + (\mathbf{D}_t + \mathbf{D}_{ts} + 3)\psi_t + V,$$

$$\psi_t = 2\left(\frac{(t-5)^4}{600} + e^{-4(t-5)^2} + \frac{t}{10} - 2\right),$$

$$U \sim \mathcal{N}(0.8V, 0.36),$$

Table 3: The data dimensions and sample splits for different datasets, where $\text{Demand}_{\text{Low}}$ denotes the Demand datasets under the low-dimensional setting, and $\text{Demand}_{\text{High}}$ denotes the Demand dataset under the high-dimensional setting.

| Setting | $\text{Dim}_X$ | $\text{Dim}_D$ | $\text{Train}_{S=1}$ | $\text{Train}_{S=0}$ | $\text{Test}_{S=1}$ | $\text{Test}_{S=0}$ |
|---|---|---|---|---|---|---|
| $\text{Demand}_{\text{Low}}$ | 2 | 10 | 4,000 | 4,000 | 1,000 | 1,000 |
| $\text{Demand}_{\text{High}}$ | 785 | 10 | 1,600 | 1,600 | 400 | 400 |
| Wage2 | 12 | 2 | 404 | 344 | 101 | 86 |

$$\mathbf{D} \sim \mathcal{N}(\mathbf{0}, \mathbf{I}),$$
$$V \sim \mathcal{N}(0, 1),$$

and

$$S = \text{Bernoulli}(\text{expit}(-T + 0.1\mathbf{D}_s + 0.1\mathbf{D}_{ts} + 0.1(X + \psi_t) + \alpha U + \beta Y)),$$

where $\text{Bernoulli}(\cdot)$ denotes the Bernoulli distribution.

### E.2    WAGE2 DATASET

The Wage2 dataset (Blackburn and Neumark, 1992) investigates the impact of years of education received ($T$) on the wage of individuals ($Y$). It includes a range of observable confounders $\mathbf{X}$ that may affect both education attainment and wage levels, such as work experience, age, residential area, marital status, and race. The IV set ($\mathbf{D}$) consists of the years of education of the individual's parents (e.g., father's education and mother's education), which influences the individual's own education ($T$) but only affects wage ($Y$) indirectly through education (Wooldridge, 2016).

To incorporate collider bias into the Wage2 dataset, we applied a non-random sampling strategy to select a subset of the original data. Individuals with higher years of education or higher wages were assigned $S = 0$. The former group (those with higher education) and the latter group (those with higher wages) may both have chosen not to report their wage outcomes, primarily due to subjective considerations such as privacy concerns or reluctance to disclose personal income-related information. Individuals' decisions on whether to disclose their wages may also be influenced by their parents' years of education, as families with higher parental education often place greater emphasis on privacy protection.

The data dimensions and sample splits for different datasets are shown in Table 3.

## F    SUPPLEMENTARY EXPERIMENTS

### F.1    EXPERIMENTS ON THE DEMAND DATASET UNDER THE HIGH-DIMENSIONAL SETTING

The data generation process for high-dimensional data is the same as that for low-dimensional data, with the selection rule defined as $\text{expit}(-T + 0.1\mathbf{D}_s + 0.1\mathbf{D}_{ts} + 0.01(\mathbf{X} + \psi) + 0.5U + 0.5Y)$. The results, as shown in Table 4, lead to the following conclusions: (1) Compared to the results on the low-dimensional data, the performance of all models significantly deteriorates due to the curse of dimensionality. (2) Heckit and IPSW struggle to learn complex nonlinear relationships, leading to significant performance degradation. (3) Deep-learning-based baselines, such as DeepGMM and DFIV, leverage neural networks to capture nonlinear relationships, resulting in superior performance in high-dimensional data. (4) DDIV achieves the best performance among all baselines, demonstrating its effectiveness and robustness in addressing coupled biases under high-dimensional conditions.

### F.2    ABLATION STUDIES

DDIV involves several important procedures and relies on various assumptions. Therefore, to assess the robustness of DDIV in situations where some procedures may not be optimal, or the assumptions may be violated, we conducted the following ablation studies on the Demand dataset with $\alpha = 1.0$ and $\beta = 1.0$.

Table 4: Out-of-sample MSE (mean $\pm$ std) on Demand datasets under the high-dimensional setting with $\alpha = 1.0$ and $\beta = 1.0$. The results in the table are scaled by a factor of $10^3$ for clarity. The best results are in bold.

| Method | Result |
|--------|--------|
| Heckit | $> 10^7$ |
| 2SRI | $1.025 \pm 0.031$ |
| IPSW | $393.8 \pm 369.1$ |
| DeepIV | $1.016 \pm 0.068$ |
| SIV | $0.856 \pm 0.112$ |
| Kernel IV | $0.958 \pm 0.035$ |
| DeepGMM | $1.022 \pm 0.051$ |
| DFIV | $1.006 \pm 0.033$ |
| CBIV | $1.047 \pm 0.046$ |
| 2SSI | $1.009 \pm 0.048$ |
| DDIV | $\mathbf{0.541 \pm 0.093}$ |

Table 5: Results from using different MI estimators on Demand datasets. The MSE results in the table are scaled by a factor of $10^3$ for clarity.

| MI Estimators | MSE | $\mathcal{I}(\Phi_t, \mathbf{D}_t)$ | $\mathcal{I}(\Phi_s, \mathbf{D}_s)$ |
|---------------|-----|-----------------|-----------------|
| NWJ | $0.305 \pm 0.055$ | 0.174 | 0.162 |
| MINE | $0.259 \pm 0.065$ | 0.241 | 0.192 |
| CLUB | $0.242 \pm 0.085$ | 0.252 | 0.179 |

### F.2.1 ABLATION STUDIES ON DECOMPOSED REPRESENTATION LEARNING

To evaluate the impact of different mutual information (MI) estimation methods, representation capacities, and hyperparameter choices on the correctness of the learned decomposed representations and the accuracy of the final estimation results, we conducted the following ablation studies:

- **Different MI estimators.** We employed three different MI estimation methods, Nguyen, Wainwright, and Jordan (NWJ) (Nguyen et al., 2010), Mutual Information Neural Estimator (MINE) (Belghazi et al., 2018), and CLUB (Cheng et al., 2020), to implement the MI minimization during the decomposed representation learning process in DDIV. During these experiments, $\lambda_1$ and $\lambda_2$ were set to $0.1$, and the hidden layer size of the representation network was set to $4$.
- **Different representation capacities.** We used different representation network architectures for decomposed representation learning, specifically varying the hidden layer sizes, including a simple linear network without a hidden layer. During these experiments, $\lambda_1$ and $\lambda_2$ were set to $0.1$.
- **Different hyperparameter choices.** We modified $\lambda_1$ and $\lambda_2$, including setting $\lambda_1 = \lambda_2 = 0$ to remove the mutual information constraint. During these experiments, the hidden layer size of the representation network was set to $4$.

We use the following metrics to evaluate the correctness of the learned decomposed representations and the accuracy of the final estimation results, respectively.

- **Correctness of the decomposed representations.** We evaluate how well the learned representations retain critical information from the IVs or SIVs by measuring the mutual information between the representations and original IVs or SIVs, i.e., $\mathcal{I}(\Phi_t, \mathbf{D}_t)$ and $\mathcal{I}(\Phi_s, \mathbf{D}_s)$. A higher value of this metric indicates better learned representations.
- **Accuracy of the estimation results.** As in the other experiments, we use out-of-sample MSE to evaluate the accuracy of the estimation results.

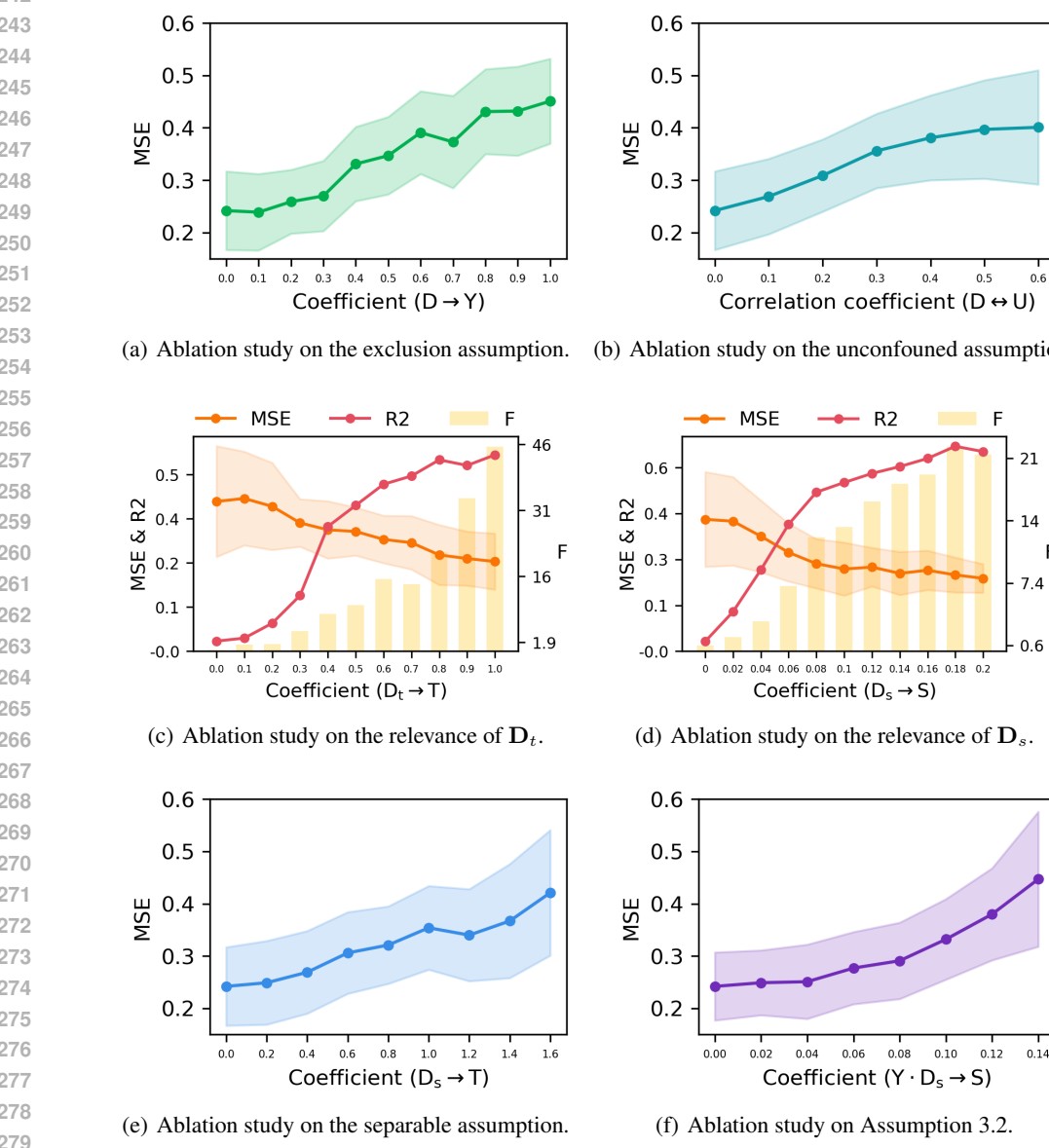

(a) Ablation study on the exclusion assumption.  (b) Ablation study on the unconfouned assumption.

(c) Ablation study on the relevance of $\mathbf{D}_t$.  (d) Ablation study on the relevance of $\mathbf{D}_s$.

(e) Ablation study on the separable assumption.  (f) Ablation study on Assumption 3.2.

Figure 5: Visualized results of ablation studies.

Table 6: Results from using different representation network architectures on Demand datasets. The MSE results in the table are scaled by a factor of $10^3$ for clarity.

| Hidden Layer Sizes | MSE | $\mathcal{I}(\Phi_t, \mathbf{D}_t)$ | $\mathcal{I}(\Phi_s, \mathbf{D}_s)$ |
|---|---|---|---|
| None | 0.262±0.062 | 0.201 | 0.169 |
| 4 | 0.242±0.085 | 0.252 | 0.179 |
| 16 | 0.249±0.087 | 0.221 | 0.183 |
| 64 | 0.243±0.081 | 0.231 | 0.186 |

We have the following observations and conclusions from the results: (1) As shown in Table 5, CLUB exhibits the best overall performance, which is why we ultimately chose CLUB as the MI estimator for DDIV. However, the performance of the other two methods, notably MINE, is also acceptable. It demonstrates that DDIV does not rely on a specific MI estimation method, highlighting its flexibility.

Table 7: Results under different hyperparameter choices on Demand datasets. The MSE results in the table are scaled by a factor of $10^3$ for clarity.

| $\lambda_1, \lambda_2$ | MSE | $\mathcal{I}(\Phi_t, \mathbf{D}_t)$ | $\mathcal{I}(\Phi_s, \mathbf{D}_s)$ |
|---|---|---|---|
| 0 | 0.303±0.092 | 0.018 | 0.005 |
| 0.001 | 0.295±0.063 | 0.052 | 0.038 |
| 0.01 | 0.313±0.098 | 0.097 | 0.065 |
| 0.1 | 0.242±0.085 | 0.252 | 0.179 |
| 1 | 0.264±0.063 | 0.234 | 0.171 |

Table 8: Out-of-sample MSE (mean ± std) results from different proportions of SIVs in the IV set. The results in the table are scaled by a factor of $10^3$ for clarity.

| Proportion | 0.0 | 0.2 | 0.5 | 0.8 |
|---|---|---|---|---|
| MSE | $0.427 \pm 0.168$ | $0.269 \pm 0.093$ | $0.242 \pm 0.085$ | $0.350 \pm 0.098$ |

(2) As shown in Table 6, the best performance is achieved with a hidden layer size of 4. However, except for the simple linear network with no hidden layers, the architecture of the representation network has little effect on the performance. It shows that DDIV is not highly sensitive to the architecture of the representation network, further demonstrating its flexibility. (3) As shown in Table 7, the best performance is achieved with a hyperparameter value of 0.1. When the hyperparameters are set too low, DDIV fails to capture sufficient information from both IVs and SIVs. On the other hand, when the hyperparameters are set too high, while the learned representation becomes more accurate, the reduced weight on the MSE loss for the treatment model may lead to worse outcome estimates. It demonstrates that DDIV is sensitive to hyperparameters, and careful selection based on validation set performance is required in practical applications.

#### F.2.2 ABLATION STUDIES ON ASSUMPTION 3.1

To evaluate the robustness of DDIV when Assumption 3.1 is not satisfied, we conducted the following ablation studies.

- **Assumption 3.1(1) is violated.** We modified the data generation process for $Y$ by incorporating a term of $\mathbf{D}$ to simulate cases where Assumption 3.1(1) is violated. We gradually increased the coefficient of this term to evaluate the performance of DDIV under varying degrees of Assumption 3.1(1) violation.
- **Assumption 3.1(2) is violated.** We modified the data generation process for $U$ and $\mathbf{D}$ as $\begin{pmatrix} U \\ \mathbf{D} \end{pmatrix} \sim \mathcal{N}\left( \begin{pmatrix} 0.8V \\ \mathbf{0} \end{pmatrix}, \begin{pmatrix} 0.36 & 0.6\boldsymbol{\gamma_0}^\top \\ 0.6\boldsymbol{\gamma_0} & \mathbf{I} \end{pmatrix} \right)$ to simulate cases where Assumption 3.1(2) is violated. We gradually increased $\boldsymbol{\gamma_0}$, the correlation coefficient between $U$ and $\mathbf{D}$, to evaluate the performance of DDIV under varying degrees of Assumption 3.1(2) violation.
- **Assumption 3.1(3) is violated.** First, we modified the coefficients of $\{\mathbf{D}_t, \mathbf{D}_{ts}\}$ and $\mathbf{D}_s$ in the data generation process for $T$ and $S$, respectively, to simulate cases where the relevance assumption for IVs and SIVs is violated. To provide a more comprehensive evaluation of the impact of weak IVs and SIVs on the correctness of decomposed representation learning, we report not only the out-of-sample MSE of the final estimates, but also the R-squared and F-statistic from the first-stage regression, where larger values are preferred. When the F-statistic is below 10, it indicates that the learned representations are too weak to satisfy the relevance assumption (Wooldridge, 2016). Second, we modified the data generation process for $T$ by incorporating a term of $\mathbf{D}_s$, and modified the proportion of $\mathbf{D}_s$ in $\mathbf{D}$, to simulate cases where SIVs are not separable from the IV set. Specifically, we employed four different settings when varying the proportion, $\{0.0, 0.2, 0.5, 0.8\}$. That is, we set the dimension of $\mathbf{D}_s$ to $\{0, 2, 5, 8\}$, with the dimensions of $\mathbf{D}_t$ and $\mathbf{D}_{ts}$ as $\{5, 4, 3, 1\}$ and

$\{5, 4, 2, 1\}$, respectively. We gradually increased the above coefficients and proportion to evaluate the performance of DDIV under varying degrees of Assumption 3.1(3) violation.

We have the following observations and conclusions from the results: (1) As shown in Figure 5(a), when the direct effect of $\mathbf{D}$ on $Y$ is small ($0.1$), indicating only a slight violation of the exclusion assumption, the performance of DDIV does not decrease. However, as this effect increases, the performance of DDIV gradually deteriorates, particularly after it exceeds $0.4$, at which point the deterioration becomes more noticeable. Nevertheless, the decrease remains smooth. The above observations demonstrate that DDIV can tolerate minor violations of the exclusion assumption. (2) As shown in Figure 5(b), as the correlation between $\mathbf{D}$ and $U$ increases, the performance of DDIV gradually deteriorates. However, this deterioration remains smooth. The above observations demonstrate that, like many other IV methods, DDIV is highly dependent on the unconfounded assumption. In practical applications, this assumption should be carefully validated through expert knowledge or sensitivity analysis. (3) As shown in Figures 5(c) and 5(d), when the coefficient of IVs is greater than $0.8$, the performance of DDIV does not show significant deterioration. However, as the relevance between the IVs and $T$ continues to decrease, the performance of DDIV gradually deteriorates, though the deterioration remains smooth. When the coefficient falls below $0.4$, the F-statistic drops below $10$, indicating a significant deterioration in the performance of DDIV. Similarly, when the correlation between the SIVs and $S$ weakens, the performance of DDIV initially does not decline, but then deteriorates smoothly as the coefficient of the SIVs falls below $0.10$. When the coefficient falls below $0.06$, the F-statistic drops below $10$, indicating a significant deterioration in the performance of DDIV. The above observations demonstrate the robustness of DDIV to weak IVs and SIVs. However, like other IV methods, it remains intolerant of IVs and SIVs that are nearly independent of $T$ and $S$, which completely violates the relevance assumption. (4) As shown in Figure 5(e), when the effect of $\mathbf{D}_s$ on $T$ is small ($0.2$), indicating only a slight violation of the separable subset assumption, the performance of DDIV does not deteriorate. As this effect increases, the performance of DDIV gradually deteriorates, but the deterioration remains smooth. The above observations demonstrate that DDIV can tolerate minor violations of the separable subset assumption. (5) As shown in Table 8, the performance of DDIV is optimal when the proportion of SIVs and IVs in the IV set is nearly equal. When the proportion of SIVs is too high, it may dilute the information provided by IVs, reducing the ability to address confounding bias. As the proportion of SIVs decreases, the performance of DDIV deteriorates smoothly and slightly. Only when the proportion reaches $0$ (i.e., the assumption is completely violated) does a significant deterioration in DDIV performance occur, demonstrating its robustness to violations of the separable subset assumption.

### F.2.3    ABLATION STUDIES ON ASSUMPTION 3.2

To evaluate the robustness of DDIV when Assumption 3.2 is not satisfied, we conducted the following ablation studies.

- **The data generation process satisfies the stricter Assumption 2.9 rather than Assumption 3.2.** We modified the data generation process for $S$ by removing the term of $U$ to simulate cases where Assumption 2.9 is satisfied.
- **The data generation process does not satisfy Assumption 3.2.** We modified the data generation process for $S$ by incorporating a multiplicative interaction term of $Y$ and $\mathbf{D}_s$ to simulate cases where Assumption 3.2 is violated. We gradually increased the coefficient of this term to evaluate the performance of DDIV under varying degrees of Assumption 3.2 violation.

We have the following observations and conclusions from the results: (1) As shown in Table 9, when Assumption 2.9 is satisfied, DDIV performs better because the biases are not coupled, proving that our method can also address non-coupled biases. The observation supports our statement that Assumption 2.9 is a special case of Assumption 3.2, demonstrating that we effectively relax the assumptions required in existing methods. (2) As shown in Figure 5(f), when the coefficient of $\mathbf{D}_s \cdot Y$ is small (below $0.06$), indicating only a slight violation of Assumption 3.2, the performance of DDIV does not deteriorate. As this coefficient increases, the performance of DDIV gradually deteriorates, but the deterioration remains smooth. The above observations demonstrate the robust of DDIV to minor violations of Assumption 3.2.

Table 9: Out-of-sample MSE (mean $\pm$ std) of DDIV under different assumptions. The results in the table are scaled by a factor of $10^3$ for clarity.

| Assumption | Result |
|---|---|
| Assumption 2.9 | $0.209 \pm 0.034$ |
| Assumption 3.2 | $0.242 \pm 0.085$ |

Table 10: ATE estimates and 95% confidence intervals (CI) under different sensitivity parameters $\gamma_1$ on the Wage2 dataset.

| IV Index | $\gamma_1$ | ATE Estimates | 95% CI |
|---|---|---|---|
| I | 0.0 | 0.074 | (0.069, 0.079) |
| I | 0.1 | 0.070 | (0.056, 0.084) |
| I | 0.2 | 0.079 | (0.065, 0.093) |
| I | 0.4 | 0.086 | (0.070, 0.102) |
| I | 0.6 | 0.102 | (0.084, 0.120) |
| II | 0.0 | 0.074 | (0.069, 0.079) |
| II | 0.1 | 0.079 | (0.073, 0.085) |
| II | 0.2 | 0.086 | (0.073, 0.099) |
| II | 0.4 | 0.074 | (0.053, 0.095) |
| II | 0.6 | 0.113 | (0.090, 0.136) |

## G  SENSITIVITY ANALYSIS

To verify whether the IV set satisfies Assumption 3.1 in the real-world dataset, we performed sensitivity analyses (Baiocchi et al., 2014) on Assumption 3.1 using the Wage2 dataset, as detailed below.

### G.1  SENSITIVITY ANALYSIS FOR VIOLATION OF ASSUMPTION 3.1(1)

This sensitivity analysis assesses whether the IV set ($\mathbf{D}$) in Wage2 satisfies the exclusion assumption, i.e., whether it does not directly affect the outcome ($Y$). To this end, we introduced a sensitivity parameter, $\gamma_1$, to represent the direct effect of $\mathbf{D}$ on $Y$, and then varied its value, transforming the original outcome into $Y - \gamma_1 \cdot T \cdot (1 - \mathbf{D})$. Then, we performed DDIV on the transformed data for the sensitivity analysis (Baiocchi et al., 2014). As reported in Table 10, the results show that as $\gamma_1$ increases, although the ATE estimates exhibit a gradually increasing bias compared to the benchmark value (0.061), and the confidence intervals also gradually expand, the magnitude of these changes remains small and slow. It demonstrates that the DDIV estimates are not sensitive to violations of Assumption 3.1(1), indicating that both IVs in Wage2 satisfy Assumption 3.1(1).

### G.2  SENSITIVITY ANALYSIS FOR VIOLATION OF ASSUMPTION 3.1(2)

This sensitivity analysis assesses whether the IV set ($\mathbf{D}$) in Wage2 satisfies the unconfounded assumption, i.e., whether it is independent of the unmeasured confounders ($\mathbf{U}$). To this end, we introduced two sensitivity parameters: (1) $\gamma_2$, the effect of $\mathbf{U}$ on the outcome $Y$; (2) $\gamma_3$, the strength of correlation between $\mathbf{U}$ and $\mathbf{D}$. By varying these parameters and transforming the original outcome into $Y - \gamma_2 \cdot \gamma_3 \cdot \mathbf{D}$, we performed DDIV on the transformed data for the sensitivity analysis (Baiocchi et al., 2014). As reported in Table 11, the results show that as $\gamma_2$ and $\gamma_3$ increases, the bias and confidence intervals of the ATE estimates grow slowly and with small magnitudes. It demonstrates

Table 11: ATE estimates and 95% confidence intervals (CI) under different sensitivity parameters $\gamma_2$ and $\gamma_3$ on the Wage2 dataset.

| IV Index | $\gamma_2$ | $\gamma_3$ | ATE Estimates | 95% CI |
|---|---|---|---|---|
| I | 0.00 | 0.00 | 0.074 | (0.069, 0.079) |
| I | 0.10 | 0.05 | 0.082 | (0.065, 0.099) |
| I | 0.10 | 0.10 | 0.090 | (0.064, 0.116) |
| I | 0.40 | 0.10 | 0.108 | (0.082, 0.134) |
| I | -0.10 | 0.10 | 0.099 | (0.073, 0.125) |
| I | -0.40 | 0.10 | 0.071 | (0.045, 0.097) |
| I | 0.10 | 0.20 | 0.116 | (0.090, 0.142) |
| II | 0.00 | 0.00 | 0.074 | (0.069, 0.079) |
| II | 0.10 | 0.05 | 0.079 | (0.065, 0.093) |
| II | 0.10 | 0.10 | 0.102 | (0.088, 0.116) |
| II | 0.40 | 0.10 | 0.099 | (0.084, 0.114) |
| II | -0.10 | 0.10 | 0.086 | (0.073, 0.099) |
| II | -0.40 | 0.10 | 0.075 | (0.059, 0.091) |
| II | 0.10 | 0.20 | 0.113 | (0.090, 0.136) |

that the DDIV estimates are not sensitive to violations of Assumption 3.1(2), indicating that both IVs in Wage2 satisfy Assumption 3.1(2).

# H    COMPUTATIONAL COMPLEXITY

Let $p_{\Phi_t}$, $p_{\Phi_s}$, $p_{\mathbf{x}}$, $p_{h_1}$, $p_{h_2}$, and $p_g$ denote the dimensions of $\Phi_t$, $\Phi_s$, $\mathbf{X}$, $h_1$, $h_2$, and $g$, respectively. $e_{\text{CLUB}}$ denotes the number of CLUB iterations used for MI estimation.

The first stage of DDIV consists of two procedures:

- Decomposed representation learning during regression for $T$ and $S$.
- Selection score estimation.

We begin by analyzing the complexity of the decomposed representation learning process. In each iteration of this process, four operations are performed simultaneously:

- Optimization of the treatment model with $O(n \cdot (p_{\Phi_t} + p_{\mathbf{x}}))$ computational complexity.
- Optimization of the selection model with $O(n \cdot (p_{\Phi_s} + p_{\mathbf{x}}))$ computational complexity.
- MI minimization between $\Phi_s$ and $T$ using CLUB (Cheng et al., 2020), with $O(e_{\text{CLUB}} \cdot n \cdot p_{\Phi_s})$ computational complexity.
- MI minimization between $\Phi_s$ and $\Phi_t$ with $O(e_{\text{CLUB}} \cdot n \cdot p_{\Phi_t} \cdot p_{\Phi_s})$ computational complexity.

Therefore, the total computational complexity of each iteration in the decomposed representation learning process is $O(n \cdot (p_{\mathbf{x}} + e_{\text{CLUB}} \cdot p_{\Phi_t} \cdot p_{\Phi_s}))$.

Next, we analyze the computational complexity of the selection score estimation process. In each iteration of this process, two operations are performed sequentially:

- Optimization of $f_{h_2}$ with $O(n \cdot p_{h_2} \cdot (p_{\Phi_s} + p_{\mathbf{x}}))$ computational complexity,.
- Optimization of $\pi$ with $O(n \cdot (p_{\Phi_s} \cdot (p_{h_1} + p_{h_2}) + p_{\mathbf{x}} \cdot (p_{h_1} + p_{h_2} + p_g)))$ computational complexity.

Therefore, the total computational complexity of each iteration in the selection score estimation process is $O(n \cdot (p_{\Phi_s} \cdot (p_{h_1} + p_{h_2}) + p_{\mathbf{x}} \cdot (p_{h_1} + p_{h_2} + p_g)))$.

Finally, the computational complexity of each iteration in the second stage of DDIV is $O(n_1 \cdot (p_{\mathbf{x}} + p_{\Phi_s}))$.

## I  BROADER IMPACTS AND ETHIC STATEMENTS

This work aims to address an unresolved challenge in causal inference, i.e., treatment effect estimation under coupled confounding and collider biases. To the best of our knowledge, it is the first work to develop identification and estimation theories for treatment effects under coupled biases, which significantly enhances the applicability of causal inference in complex real-world scenarios.

We do not foresee any negative societal impacts. The research does not involve sensitive data, high-risk models, or ethically sensitive applications, and is intended to enhance fairness and robustness in data-driven decision-making.

## J  LLM USAGE STATEMENT

LLMs were only used to polish writing. They offered suggestions on grammar, vocabulary, and other minor adjustments. We carefully reviewed and implemented relevant suggestions to improve the readability of the manuscript.

