# OpenReview forum: "Identification and Estimation of Treatment Effects under Coupled Confounding and Collider Biases"
_ICLR.cc/2026/Conference — Submitted to ICLR 2026_

### Official Review · Reviewer_hNeH · 2025-10-21

**Soundness:** 3
**Presentation:** 3
**Contribution:** 4
**Rating:** 6
**Confidence:** 4

**Summary:**

The paper studies treatment-effect estimation when two biases act together: unmeasured confounding and collider (selection) bias. The authors assume access to an "IV set" $D$ that contains variables influencing treatment and selection separately, with exclusion and independence properties. They prove an identification result for $E[Y \mid d o(T=t), X]$ under a separable selection model that allows unmeasured factors to affect selection. They then propose DualDebiasIV (DDIV), a two-stage estimator that (i) decomposes the IV set into an IV-like representation $\Phi_t$ and a selection-IV representation $\Phi_s$ via mutual-information constraints, fits $T$ on ( $X, \Phi_t$ ), and learns the selection score $\pi\left(X, T, Y, \Phi_s\right)$; and (ii) reweights by $\pi^{-1}$ and regresses $Y$ on $\left(X, \hat{T}, \Phi_s\right)$. The paper presents consistency arguments and experiments on a semi-synthetic demand benchmark and the Wage2 dataset, showing lower MSE and ATE estimates closer to a reference value.

**Strengths:**

Originality.
- The paper targets the coupled-bias setting explicitly and gives an identification theorem that combines an IV subset for treatment and a selection-IV subset for sampling. This is a new blend of ideas that is not handled by standard IV or selection models alone. The decomposition of an IV set into $\Phi_t$ and $\Phi_s$ with MI penalties is also novel as a practical mechanism to operationalize the theory.

Quality.
- The identification section states clear structural assumptions (IV-set exclusion and independence; separable selection with unobservables), and the estimation section aligns with those assumptions.
- The algorithm is consistent with the theory: first remove selection bias with $\Phi_s$ and $\pi$, then address confounding with $\Phi_t$ (via fitted $\hat{T}$ ).

Clarity.
- The problem is motivated with graphs and a real-world story. The assumptions are labeled, and the two-stage pipeline is easy to follow.
- Limitations are acknowledged (need for an IV set; tuning and architecture choices).

Significance.
- Many ML+causal settings face both residual confounding and non-random selection. A framework that unifies identification and estimation for this case can influence practice across economics, health, and policy analytics.
- The approach is modular: it can plug into representation learners and outcome models, making it relevant to ICLR.

**Weaknesses:**

1) Assumptions and their testability.

The main identification rests on three strong pieces:
- (a) a supplied IV set $D$ that is independent of $(X, U)$ and excluded from $Y$;
- (b) a known split of $D$ into $D_t$ (for confounding) and $D_s$ (for selection);
- (c) an additively separable selection model.

Assumption (b) is especially demanding in practice. It asks the user to know, in advance, which elements of $D$ shift treatment only ($D_t\not\perp T\mid X$) and which shift selection only ($D_s\not\perp S\mid (X,T); D_s\perp T$). This is not clear in real data how to determine $D_s$ from $D_t$. Also, the paper should provide tools and evidence that make (a) and (b) more credible and more robust, e.g., provide a sensitivity analysis that perturbs (i) exclusion in $D$ and (ii) separability in selection, and report how bias/variance change.

2. Role and discovery of the IV set.
- The method presumes a supplied $D$ with known $D_t$ and $D_s$. In many ML applications, $D$ is high-dimensional and noisy, with some components weak or invalid. Please add experiments that (a) inject invalid instruments into $D$ (direct paths to $Y$ or correlation with $U$ ) and (b) vary instrument strength and overlap. The authors can clarify when $\Phi_t, \Phi_s$ remain valid under various contaminations.

3. Mutual-information penalties and identifiability in practice.
- Proposition 4.1 hinges on "sufficiently rich models," consistent MI estimation, and large penalty weights. In finite samples, CLUB can be biased and high-variance. Can the authors add ablations over MI estimators (CLUB vs. NWJ vs. MINE), penalty magnitudes, and representation capacity?

4. Target estimand and heterogeneity.
- The identification section mixes conditional effects $E[Y \mid do(T=t), X]$ and ATE in experiments. Please be explicit about the target estimand for each experiment and about if the homogeneity assumption (Assumption 2.4) is needed to link IV identification to that estimand. Another question is that how does general effect heterogeneity (general CATE) affect the validity of Assumption 2.4? In other words, can you give an example of data generating models with general CATEs which simultaneously satisfies Assumptions 2.1, 2.2, 2.4, 3.1, and 3.2? If the target causal estimand is ATE, can the authors provide valid confidence intervals?

**Questions:**

1. Diagnosing the IV set.

How should a user check that the supplied $D$ is plausibly independent of ($X, U$) and satisfies exclusion with respect to $Y$ ? Can you propose empirical proxies or falsification tests that your pipeline can output by default?

2. When $D$ is partially invalid.

If a fraction of variables in $D$ violate exclusion or independence, do the MI constraints still recover usable $\Phi_t$ and $\Phi_s$ ? Please provide theory or a robustness experiment that quantifies the breakdown point.

3. Separable selection model.

Your identification needs logit$P\left(S=1 \mid U, X, T, D_s, Y\right)=q\left(X, T, D_s\right)+h(U, X, T, Y)$. How sensitive are results to mild non-separability (e.g., small $D_s \times Y$ interactions)? Can you add a sensitivity curve that sweeps an interaction term from 0 to moderate values?

4. Instrument strength and overlap.

What happens as $D_t$ becomes weak or as overlap in $P(T \mid X)$ or $P(S \mid X)$ worsens? Please report performance vs. first-stage $R^2$, F-statistics, and minimum/maximum selection propensities, including failure thresholds.

5. Computational cost.

What is the training time relative to DeepIV/DeepGMM on the same hardware? Include wall-clock and parameter counts; discuss scalability to large $D$ or image/text covariates.

---

> ### Author Response · Authors · 2025-12-03
>
> We sincerely appreciate the reviewer’s great efforts and insightful comments to improve our manuscript. In below, we address these concerns point by point.
>
> > **[W1.1] How to determine $\mathbf{D}_s$ from $\mathbf{D}_t$.**
>
> We thank the reviewer for the insightful suggestion. **We can determine $\mathbf{D}_s$ from $\mathbf{D}_t$ by directly calculating the correlation coefficients between each IV and $T$, and between each IV and $S$. That is, if an IV is correlated with $T$, it belongs to $\mathbf{D}_t$; if an IV is independent of $T$ and correlated with $S$, it belongs to $\mathbf{D}_s$.** We thank the reviewer again and have included the corresponding discussions in the revised manuscript (lines 208-212).
>
> > **[W1.2&Q1] How to check whether Assumption 3.1(1) and Assumption 3.1(2) are satisified.**
>
> We thank the reviewer for the valuable suggestions. **Assumptions 3.1(1) and (2) can be validated through sensitivity analysis [1]. Following these suggestions, we have also performed sensitivity analysis on the real-world dataset, Wage2, as an illustrative procedure for checking these assumptions, detailed in Appendix G of the revised manuscript.**
>
> > **[W2.1&Q2&Q3&Q4.1] Ablation studies on Assumptions 3.1 and 3.2.**
>
> We thank the reviewer for the insightful suggestions. **Following the suggestions, we have conducted extensive ablation studies for all the scenarios mentioned, where Assumptions 3.1 and 3.2 are violated, as detailed in Appendix F.2.2 and Appendix F.2.3 of the revised manuscript.** The results demonstrate that DDIV is robust to minor violations of these assumptions.
>
> > **[W3] Ablation studies on decomposed representation learning.**
>
> We thank the reviewer for the insightful suggestion. **Following the suggestion, we have conducted extensive ablation studies decomposed representation learning, including varying hyperparameters, representation capacities, and MI estimators, detailed in Appendix F.2.1 of the revised manuscript.** The results show that DDIV is not highly sensitive to the specific MI estimation method used and the architecture of the representation network, demonstrating its flexibility. However, DDIV is sensitive to hyperparameters, and careful selection based on validation set performance is required in practical applications.
>
> > **[W4] Target estimand and heterogeneity.**
>
> We thank the reviewer for the thoughtful comments. We would like to clarify these questions as follows.
>
> (1) The target estimands for each dataset are as follows: 1. **The target estimand for Demand is $\mathbb{E}[Y\mid do(T=t),\mathbf{X}]$**, as it is a semi-synthetic dataset with continuous treatments and ground-truth counterfactual outcomes. 2. **The target estimand for Wage2 is ATE**, since it is a real-world dataset with binary treatments and lacks ground-truth counterfactual outcomes.
>
> (2) **Our theory covers the identification of $\mathbb{E}[Y\mid do(T=t),\mathbf{X}]$, CATE, and ATE.** It is possible that the absence of CATE estimation experiments in the original manuscript led to some misunderstanding by the reviewer. **Therefore, we followed [2] and conducted additional experiments on a semi-synthetic benchmark for CATE estimation, IHDP, with binary treatments and ground-truth CATEs.** Specifically, we generated a $3$-dimensional IV vector $\{D_1, D_2, D_3\}$ following a multivariate normal distribution: $\{D_1, D_2, D_3\} \sim \mathcal{N}(\mathbf{0}, \mathbf{I})$. The data generation process for $Y$ and $T$ followed [2], and we introduced selection bias into it through $S \sim \text{Bernoulli}\left(\frac{1}{1 + \exp\left(Y + T + 0.5 * D_3 + 0.2 * D_1 + 0.1 * \left(\sum X_i + \sum U_i\right)\right)}\right)$. Following [3], we report the mean and standard deviation of $\sqrt{PEHE}$, where smaller values indicate better performance.
>
> | Method    | $\sqrt{PEHE}$  |
> | --------- | --------------------- |
> | Heckit    | 3.723$\pm$0.450       |
> | 2SRI      | 3.149$\pm$0.344       |
> | IPSW      | 2.204$\pm$0.055       |
> | DeepIV    | 2.105$\pm$0.403       |
> | SIV       | 4.039$\pm$0.010       |
> | Kernel IV | 3.512$\pm$0.214       |
> | DeepGMM   | 2.846$\pm$0.464       |
> | DFIV      | 3.713$\pm$0.240       |
> | CBIV      | 3.203$\pm$0.010       |
> | 2SSI      | 2.871$\pm$0.345       |
> | **DDIV**      | **1.411$\pm$0.141**  |
>
> **The results show that DDIV also outperforms all baselines and achieves optimal performance in CATE estimation.**
>
> (3) **Heterogeneity of treatment effects on $\mathbf{X}$ does not violate Assumption 2.4, as Assumption 2.4 only prohibits heterogeneity in treatment effects on $\mathbf{U}$.** An example of data generating models that satisfies all the required assumptions of DDIV is $Y = X \cdot T + U$; $T = X \cdot D_t + U$; $\mathrm{logit}(\mathbb{P}(S=1\mid U,X,T,Y,D_s))=X\cdot T\cdot D_s + U \cdot X \cdot T \cdot Y$.
>
> (4) DDIV's 95\% confidence interval for the ATE estimate on the Wage2 dataset is $(0.069, 0.079)$.

---

> ### Author Response · Authors · 2025-12-03
>
> > **[W2.2&Q4.2] Ablation studies on the overlap assumption.**
>
> We thank the reviewer for the insightful suggestions. Due to the difficulty in controlling the degree of overlap in the continuous treatments of the semi-synthetic dataset (Demand) in the original manuscript, **we conducted ablation studies on the overlap assumption using the IHDP dataset.** We adjusted the coefficients of $\mathbf{X}$ in the data generation processes for $T$ and $S$ to evaluate the performance of DDIV under varying degrees of overlap violation.
>
> | $\sqrt{PEHE}$  | $R^2$    | $F$     | $\mathbb{P}(T\mid \mathbf{X})$ maximum | $\mathbb{P}(T\mid \mathbf{X})$ minimum |
> | --------------- | ----- | ----- | ----- | ----- |
> | 1.426$\pm$0.212 | 0.372 | 9.135 | 0.618 | 0.409 |
> | 1.407$\pm$0.315 | 0.582 | 14.62 | 0.911 | 0.085 |
> | 1.209$\pm$0.214 | 0.526 | 12.41 | 0.927 | 0.018 |
> | 1.526$\pm$0.426 | 0.416 | 8.626 | 0.977 | 0.009 |
> | 1.914$\pm$0.512 | 0.257 | 4.726 | 0.999 | 0.001 |
>
> | $\sqrt{PEHE}$   | $R^2$    | $F$     | $\mathbb{P}(S\mid \mathbf{X})$ maximum | $\mathbb{P}(S\mid \mathbf{X})$ minimum |
> | --------------- | ----- | ----- | ----- | ----- |
> | 0.813$\pm$0.105 | 0.474 | 18.51 | 0.629 | 0.351 |
> | 1.041$\pm$0.562 | 0.692 | 26.13 | 0.973 | 0.034 |
> | 1.209$\pm$0.214 | 0.416 | 13.63 | 0.941 | 0.008 |
> | 2.204$\pm$0.415 | 0.155 | 3.961 | 0.994 | 0.003 |
> | 3.415$\pm$1.515 | 0.104 | 1.425 | 0.999 | 0.001 |
>
> **The results show that DDIV can tolerate a significant degree of insufficient overlap.** However, when the overlap assumption is nearly completely violated (i.e., when the propensity minimum is below $0.01$), both the accuracy of the final result estimates ($\sqrt{PEHE}$ increases sharply) and the correctness of the learned representations ($F<10$ [1]) are significantly reduced. **The above obsevations demonstrate that DDIV is robust to violations of the overlap assumption in non-extreme cases.**
>
>
> > **[W5] Computational cost.**
>
> We thank the reviewer for the valuable suggestion. **We have included a computational complexity analysis in Appendix H of the revised manuscript.** Let $p_{\Phi_t}$, $p_{\Phi_s}$, $p_\mathbf{x}$, $p_{h_1}$, $p_{h_2}$, and $p_{g}$ denote the dimensions of $\Phi_t$, $\Phi_s$, $\mathbf{X}$, $h_1$, $h_2$, and $g$, respectively. $e_{\mathrm{CLUB}}$ denotes the number of CLUB iterations used for MI estimation. **Briefly, the DDIV consists of three sequential procedures: (1) Decomposed representation learning of Stage 1, which has a time complexity of $O(n\cdot(p_\mathbf{x}+  e_{\mathrm{CLUB}}\cdot p_{\Phi_t}\cdot p_{\Phi_s}))$ per iteration; (2) selection score estimation of Stage 1, with a time complexity of $O(n\cdot(p_{\Phi_s} \cdot (p_{h_1}+p_{h_2})+p_\mathbf{x}\cdot (p_{h_1}+p_{h_2}+p_g)))$ per iteration; (3) Stage 2 has a time complexity of $O(n_1 \cdot (p_\mathbf{x}+p_{\Phi_s}))$ per iteration.** The detailed analysis can be found in Appendix H.
>
> Compared to other IV methods based on representation learning (e.g., DFIV with per-iteration time complexity of $O(n \cdot (p_{\mathbf{x}}+p_{\Phi_t}))$ for Stage 1 and $O(n_1 \cdot p_{\mathbf{x}})$ for Stage 2), DDIV has a slightly higher time complexity. This increase is primarily due to: (1) the mutual information estimation in the first stage, which adds $O(e_{\mathrm{CLUB}}\cdot p_{\Phi_t}\cdot p_{\Phi_s})$; (2) the selection score estimation between the first and second stages, which adds $O(n\cdot(p_{\Phi_s} \cdot (p_{h_1}+p_{h_2})+p_\mathbf{x}\cdot (p_{h_1}+p_{h_2}+p_g)))$.
>
> We also provide a runtime comparison of each method on Demand, as shown in the table below.
>
> | Method | Runtime (s) |
> | :-: | :-: |
> | Heckit | 0.10 |
> | 2SRI | 0.11 |
> | IPSW | 0.08 |
> | DeepIV | 108. |
> | SIV | 6.23 |
> | Kernel IV | 8.49 |
> | DeepGMM | 179. |
> | DFIV | 5.31 |
> | CBIV | 1.71 |
> | 2SSI | 6.85 |
> | DDIV | 8.44 |
>
> As shown, while DDIV has a longer running time compared to other methods based on representation learning, such as DFIV and 2SSI, the difference is not significant. Moreover, it is still significantly faster than methods like DeepIV and DeepGMM. **Therefore, although the time complexity of DDIV is not optimal, it is still considered acceptable.**
>
> When dealing with high-dimensional data such as image/text, using these high-dimensional features as covariates does not significantly increase the computational cost, as the computational complexity of DDIV with respect to $p_\mathbf{x}$ is mostly linear. However, suppose these high-dimensional features are used as IVs, the computational cost may be quite high, as DDIV has quadratic complexity with respect to $p_{\Phi_t}$ and $p_{\Phi_s}$.

---

> > ### Author Response · Authors · 2025-12-03
> >
> > **We hope the above discussion will fully address the reviewer's concerns about our work, and we would really appreciate it if the AC could consider this response and recommend its acceptance for publication. We would like to express our gratitude to both the reviewer and the AC for their great efforts and constructive comments on our manuscript. Thank you!**
> >
> > > **References**
> >
> > [1] Wooldridge, J. M. (2016). Introductory econometrics a modern approach. South-Western cengage learning.
> >
> > [2] Wu, A., Kuang, K., Li, B., & Wu, F. (2022, June). Instrumental variable regression with confounder balancing. In International Conference on Machine Learning (pp. 24056-24075). PMLR.
> >
> > [3] Shalit, U., Johansson, F. D., & Sontag, D. (2017, July). Estimating individual treatment effect: generalization bounds and algorithms. In International conference on machine learning (pp. 3076-3085). PMLR.

---

### Official Review · Reviewer_y1LA · 2025-10-31

**Soundness:** 3
**Presentation:** 3
**Contribution:** 2
**Rating:** 4
**Confidence:** 4

**Summary:**

The paper studies the problem of identifying and estimating treatment effects when both confounding bias and collider bias exist and are coupled, meaning that unobserved confounders influence both treatment assignment and sample selection. The authors propose a new identification theory using an instrumental variable (IV) set that includes subsets serving as both IVs and selection IVs (SIVs). Based on this theory, they develop DualDebiasIV (DDIV), a two-stage estimation approach that decomposes the IV set into IV and SIV representations using mutual information constraints, and then corrects for coupled biases through reweighting and regression. The paper provides proofs of identification and consistency, and experimental results on semi-synthetic and real datasets show that DDIV outperforms existing methods.

**Strengths:**

1. The paper is well written and easy to follow. The motivation and problem setup are clearly presented, with intuitive examples and well-structured explanations.

1. The problem studied is both interesting and important, addressing a setting rarely discussed in causal inference.

1. This paper introduces a method that decomposes IVs into IV and SIV representations via mutual information minimization.

**Weaknesses:**

1. The claim that “the coupled bias problem remains unresolved” is somewhat misleading, as it has been studied in [1]. The paper should better position its contribution by explicitly discussing how it differs from or extends prior work.
1. It is unclear whether inference stages can be conducted to establish the asymptotically normal property. A discussion on this aspect would strengthen the theoretical contribution.
1. It is clear that $D_t \not \perp T$ and $D_s \perp T$. Thus, they are easily separated by independence test. Why do the authors rely on representation learning, which may introduce inaccuracy due to model complexity and the difficulty of mutual information estimation?
1. The reason for splitting the data (Batch 1 and Batch 2(A)) in Algorithm 1 is not explained. Clarifying the motivation and necessity of this step would help readers understand the algorithmic design.
1. Real-world validation is limited to one dataset where coupled biases are artificially introduced, reducing the practical credibility of the results.



[1] Li, Baohong, et al. "Two-stage shadow inclusion estimation: an IV approach for causal inference under latent confounding and collider bias." *Forty-first International Conference on Machine Learning*. 2024.

**Questions:**

see above

---

> ### Author Response · Authors · 2025-12-03
>
> We sincerely appreciate the reviewer’s great efforts and insightful comments to improve our manuscript. In below, we address these concerns point by point.
>
> > **[W1] How DDIV differs from 2SSI [1].**
>
> We thank the reviewer for the insightful suggestion. As stated in their paper, **2SSI assumes that the unmeasured confounder is conditionally independent of $S$, i.e., $U \not \rightarrow S$.** Therefore, it only addresses two INDEPENDENT confounding and collider biases that coexist simultaneously and **cannot address coupled biases, where $U \rightarrow S$.** We thank the reviewer again and have included a relevant discussion on this point in Appendix A of the revised manuscript.
>
> > **[W2] Discussions on the asymptotically normal property of DDIV.**
>
> We thank the reviewer for the valuable suggestion. **According to our Theorem 3.4, under the condition of correct model specification, the unbiasedness of the estimator is theoretically guaranteed. Therefore, under the conditions specified in previous work [2-4], both consistency and asymptotic normality of DDIV are also theoretically guaranteed.** Of course, in practical applications, these conditions are not always perfectly satisfied, as they are influenced by factors such as the architecture of the representation network and training hyperparameters. **Therefore, we have also conducted relevant ablation studies on the Demand dataset, detailed in the response to [W3].**
>
> The detailed proof of Theorem 3.4 is in Appendix B.3, and more details on the supplementary ablations are in Appendix F.2.1 of the revised manuscript.
>
> > **[W3] Implementation of DDIV using decomposed representation learning versus variable selection.**
>
> We thank the reviewer for the insightful suggestion. Our theory of identification and estimation can also be extended to a variable selection framework, as the reviewer suggests. **The reason we employ decomposed representation learning in our implementation is to enhance its flexibility when dealing with high-dimensional data. To further address the reviewer’s concerns regarding the accuracy of decomposed representation learning, we have conducted extensive ablation studies, detailed below.**
>
> 1. **Different MI estimators.** We employed three different MI estimation methods, Nguyen, Wainwright, and Jordan (NWJ) [5], Mutual Information Neural Estimator (MINE) [6], and CLUB [7], to implement the MI minimization during the decomposed representation learning process in DDIV. During these experiments, $\lambda_1$ and $\lambda_2$ were set to $0.1$, and the hidden layer size of the representation network was set to $4$.
> 2. **Different representation architectures.** We used different representation network architectures for decomposed representation learning, specifically varying the hidden layer sizes, including a simple linear network without a hidden layer. During these experiments, $\lambda_1$ and $\lambda_2$ were set to $0.1$.
> 3. **Different hyperparameter choices.** We modified the hyperparameters $\lambda_1$ and $\lambda_2$. During these experiments, the hidden layer size of the representation network was set to $4$.
>
> Except for the out-of-sample MSE, we also report the mutual information between the representations and original IVs or SIVs, i.e., $\mathcal{I}(\Phi_t, \mathbf{D}_t)$ and $\mathcal{I}(\Phi_s, \mathbf{D}_s)$, to measure how well the learned representations retain critical information from the IVs or SIVs. A higher value of this metric indicates better learned representations.
>
> | MI Estimators | MSE  | $\mathcal{I}(\Phi_t, \mathbf{D}_t)$ | $\mathcal{I}(\Phi_s, \mathbf{D}_s)$ |
> | --- | --- | --- | --- |
> | NWJ | 0.305$\pm$0.055 | 0.174 | 0.162 |
> | MINE | 0.259$\pm$0.065 | 0.241 | 0.192 |
> | CLUB  | 0.242$\pm$0.085 | 0.252 | 0.179 |
>
> | Hidden Layer Sizes | MSE    | $\mathcal{I}(\Phi_t, \mathbf{D}_t)$ | $\mathcal{I}(\Phi_s, \mathbf{D}_s)$ |
> | --- | ---| --- | --- |
> | None   |  0.262$\pm$0.062 | 0.201 | 0.169  |
> | 4     | 0.242$\pm$0.085 | 0.252 | 0.179 |
> | 16   | 0.249$\pm$0.087 | 0.221 | 0.183 |
> | 64   | 0.243$\pm$0.081 | 0.231 | 0.186 |
>
> | Hyperparameters | MSE     | $\mathcal{I}(\Phi_t, \mathbf{D}_t)$ | $\mathcal{I}(\Phi_s, \mathbf{D}_s)$ |
> | ---------- | -------- | ------------------------------------- | ------------------------------------- |
> | 0      | 0.303$\pm$0.092 | 0.018 | 0.005    |
> | 0.001 | 0.295$\pm$0.063 | 0.052 | 0.038 |
> | 0.01 | 0.313$\pm$0.098 | 0.097 | 0.065 |
> | 0.1 | 0.242$\pm$0.085 | 0.252 | 0.179 |
> | 1 | 0.264$\pm$0.063 | 0.234 | 0.171 |
>
> **The results show that DDIV is not highly sensitive to the specific MI estimation method used and the architecture of the representation network, demonstrating its flexibility. However, DDIV is sensitive to hyperparameters, and careful selection based on validation set performance is required in practical applications.**
>
> We thank the reviewer again and have included the above ablations with more detailed analysis in Appendix F.2.1 of the revised manuscript.

---

> > ### Author Response · Authors · 2025-12-03
> >
> > > **[W4] The reason for splitting the data (Batch 1 and Batch 2(A)) in Algorithm 1.**
> >
> > We thank the reviewer for the valuable suggestion. **The reason we split the data and used different splits at different stages is to avoid overfitting, following DML [8].** We would also like to clarify the meaning of "Batch 2(A)" in Algorithm 1. In the original manuscript, "Batch 2(A)" refers to the samples with $S=1$, and the corresponding "Batch 2(B)" refers to the samples with $S=0$. The distinction is made because only the samples with $S=1$ have available outcome values. Therefore, during the second stage, we use only this subset of Batch 2 for training, while the remaining subset ($S=0$) is used solely for testing. We thank the reviewer again and have revised the pseudocode to clarify these potentially confusing parts.
> >
> > > **[W5] Experiments on more real-world datasets.**
> >
> > We thank the reviewer for the valuable suggestion. Following this suggestion, **we have conducted additional experiments on three more datasets:**
> >
> > (1) **The semi-synthetic dataset IHDP, which evaluates the performance of DDIV for CATE estimation.** We generated a $3$-dimensional IV vector $\{D_1, D_2, D_3\}$ following a multivariate normal distribution: $\{D_1, D_2, D_3\} \sim \mathcal{N}(\mathbf{0}, \mathbf{I})$. The data generation process for $Y$ and $T$ followed [9], and we introduced selection bias into it through $S \sim \text{Bernoulli}\left(\frac{1}{1 + \exp\left(Y + T + 0.5 * D_3 + 0.2 * D_1 + 0.1 * \left(\sum X_i + \sum U_i\right)\right)}\right)$. Following [10], we report the mean and standard deviation of $\sqrt{PEHE}$, where smaller values indicate better performance.
> > | Method    | $\sqrt{PEHE}$  |
> > | --------- | --------------------- |
> > | Heckit    | 3.723$\pm$0.450       |
> > | 2SRI      | 3.149$\pm$0.344       |
> > | IPSW      | 2.204$\pm$0.055       |
> > | DeepIV    | 2.105$\pm$0.403       |
> > | SIV       | 4.039$\pm$0.010       |
> > | Kernel IV | 3.512$\pm$0.214       |
> > | DeepGMM   | 2.846$\pm$0.464       |
> > | DFIV      | 3.713$\pm$0.240       |
> > | CBIV      | 3.203$\pm$0.010       |
> > | 2SSI      | 2.871$\pm$0.345       |
> > | **DDIV**      | **1.411$\pm$0.141**       |
> >
> > (2, 3) **The real-world SMOKE dataset [11] and CARD dataset [12], which evaluate the performance of DDIV for ATE estimation in more diverse real-world scenarios.** Similar to the Wage2 dataset, we introduced selection bias by applying a non-random sampling strategy to select a subset of the original data. Specifically, we set $S=1$ for those with higher values of SIV and $T$ and lower values of $Y$. We use the results from the previous study [5, 6] as benchmark results and compare the estimated ATEs with them to evaluate the accuracy of ATE estimation, i.e., methods with results closer to the benchmark results perform better.
> >
> > Results on the SMOKE dataset:
> >
> > |  Method    | ATE$\pm$95\%CI |
> > | --------- | ---------------- |
> > | Heckit    | 0.003$\pm$0.001  |
> > | 2SRI      | 0.011$\pm$0.005  |
> > | IPSW      | -0.116$\pm$0.052 |
> > | DeepIV    | 0.000$\pm$0.001  |
> > | SIV       | -0.079$\pm$0.038 |
> > | Kernel IV | 0.117$\pm$0.54   |
> > | DeepGMM   | 0.002$\pm$0.001  |
> > | DFIV      | -0.238$\pm$0.062 |
> > | CBIV      | -0.013$\pm$0.110 |
> > | 2SSI      | -0.087$\pm$0.035 |
> > | **DDIV**      | **-0.025$\pm$0.016** |
> > | Reference | -0.042$\pm$0.026 |
> >
> > Results on the CARD dataset:
> >
> > |  Method   | ATE$\pm$95\% CI |
> > | --------- | ---------------- |
> > | Heckit    | 0.002$\pm$0.001  |
> > | 2SRI      | 0.007$\pm$0.003  |
> > | IPSW      | -0.016$\pm$0.007 |
> > | DeepIV    | 0.006$\pm$0.001  |
> > | SIV       | 0.024$\pm$0.010  |
> > | Kernel IV | -0.351$\pm$0.114 |
> > | DeepGMM   | -0.003$\pm$0.001 |
> > | DFIV      | 0.529$\pm$0.094  |
> > | CBIV      | 0.643$\pm$0.510  |
> > | 2SSI      | 0.167$\pm$0.105  |
> > | **DDIV**      | **0.091$\pm$0.036**  |
> > | Reference | 0.117$\pm$0.047  |
> >
> >
> > **From the results, it is evident that DDIV outperforms all baselines and achieves optimal performance across all three datasets, demonstrating its effectiveness in addressing various real-world challenges.**
> >
> > ***
> > **We hope the above discussion will fully address the reviewer's concerns about our work, and we would really appreciate it if the AC could consider this response and recommend its acceptance for publication. We would like to express our gratitude to both the reviewer and the AC for their great efforts and constructive comments on our manuscript. Thank you!**

---

> > > ### Author Response · Authors · 2025-12-03
> > >
> > > > **References**
> > >
> > > [1] Li, B., Wu, A., Xiong, R., & Kuang, K. (2024, July). Two-stage shadow inclusion estimation: an IV approach for causal inference under latent confounding and collider bias. In Forty-first International Conference on Machine Learning.
> > >
> > > [2] Newey, W. K., & McFadden, D. (1994). Large sample estimation and hypothesis testing. Handbook of econometrics, 4, 2111-2245.
> > >
> > > [3] Newey, W. K., & Powell, J. L. (2003). Instrumental variable estimation of nonparametric models. Econometrica, 71(5), 1565-1578.
> > >
> > > [4] Sun, B., Liu, L., Miao, W., Wirth, K., Robins, J., & Tchetgen, E. J. T. (2018). Semiparametric estimation with data missing not at random using an instrumental variable. Statistica Sinica, 28(4), 1965.
> > >
> > > [5] Nguyen, X., Wainwright, M. J., & Jordan, M. I. (2010). Estimating divergence functionals and the likelihood ratio by convex risk minimization. IEEE Transactions on Information Theory, 56(11), 5847-5861.
> > >
> > > [6] Belghazi, M. I., Baratin, A., Rajeshwar, S., Ozair, S., Bengio, Y., Courville, A., & Hjelm, D. (2018, July). Mutual information neural estimation. In International conference on machine learning (pp. 531-540). PMLR.
> > >
> > > [7] Cheng, P., Hao, W., Dai, S., Liu, J., Gan, Z., & Carin, L. (2020, November). Club: A contrastive log-ratio upper bound of mutual information. In International conference on machine learning (pp. 1779-1788). PMLR.
> > >
> > > [8] Chernozhukov, V., Chetverikov, D., Demirer, M., Duflo, E., Hansen, C., Newey, W., & Robins, J. (2018). Double/debiased machine learning for treatment and structural parameters.
> > >
> > > [9] Wu, A., Kuang, K., Li, B., & Wu, F. (2022, June). Instrumental variable regression with confounder balancing. In International Conference on Machine Learning (pp. 24056-24075). PMLR.
> > >
> > > [10] Shalit, U., Johansson, F. D., & Sontag, D. (2017, July). Estimating individual treatment effect: generalization bounds and algorithms. In International conference on machine learning (pp. 3076-3085). PMLR.
> > >
> > > [11] Wooldridge, J. M. (2016). Introductory econometrics a modern approach. South-Western cengage learning.
> > >
> > > [12] Card, D. (1993). Using geographic variation in college proximity to estimate the return to schooling.

---

### Official Review · Reviewer_iYvR · 2025-11-01

**Soundness:** 2
**Presentation:** 3
**Contribution:** 3
**Rating:** 4
**Confidence:** 3

**Summary:**

This paper claims to address a long-standing challenge in causal inference for observational studies: coupled confounding and collider biases. To solve this, this paper proposes an identification theory for treatment effects under coupled biases, which relies on a strong assumption that a known IV set containing subsets that act as IV and SIV respectively. This paper validates the method on semi-synthetic  and real-world datasets, showing it outperforms baselines.

**Strengths:**

1. The paper’s identification theory is novel.
2. The paper meets good standards of quality across theoretical, methodological, and experimental dimensions.
3. The paper is well-organized and accessible to both causal inference experts and researchers familiar with machine learning.

**Weaknesses:**

1. The paper’s foundational Assumption 3.1 (“Known Instrumental Variable Set”) requires a pre-identified set containing disjoint IV and SIV subsets. This assumption is unrealistic for most real-world scenarios.
2. The paper claims to be the “first work to address coupled confounding and collider biases”, but this is misleading. Prior studies such as [1] already tackle joint bias correction.
3. The paper’s experiments are well-controlled but fail to validate DDIV’s performance in scenarios that reflect real-world challenges.

[1] Li, B., Wu, A., Xiong, R. &amp; Kuang, K.. (2024). Two-Stage Shadow Inclusion Estimation: An IV Approach for Causal Inference under Latent Confounding and Collider Bias. <i>Proceedings of the 41st International Conference on Machine Learning</i>, in <i>Proceedings of Machine Learning Research</i> 235:28949-28964 Available from https://proceedings.mlr.press/v235/li24bu.html.

**Questions:**

See above.

---

> ### Author Response · Authors · 2025-12-03
>
> We sincerely appreciate the reviewer’s great efforts and insightful comments to improve our manuscript. In below, we address these concerns point by point.
>
> > **[W1] How realistic Assumption 3.1 is.**
>
> We thank the reviewer for the thoughtful comment. However, we respectfully disagree with the statement that  "Assumption 3.1 is unrealistic for most real-world scenarios." While Assumption 3.1 may not always hold in the real world, **it is satisfied in many scenarios, particularly when some weak IVs are available in the IV set, which naturally satisfy all the conditions of Assumption 3.1 if some of these weak IVs also influence sample selection.** We also provide an example in the manuscript (Figure 2) where Assumption 3.1 holds. Moreover, **we would like to clarify that Assumption 3.1 is, to some extent, testable in practical applications.** Specifically, Assumptions 3.1(1) and (2) can be validated through sensitivity analysis [1], as detailed in Appendix G of the revised manuscript. As for Assumption 3.1(3), it can also be verified by directly calculating the correlation coefficients between each IV and $T$, and between each IV and $S$. That is, if an IV is correlated with $T$, it belongs to $\mathbf{D}_t$; if an IV is independent of $T$ and correlated with $S$, it belongs to $\mathbf{D}_s$. We have also performed sensitivity analysis on the real-world dataset, Wage2, as an illustrative procedure for testing Assumption 3.1, detailed in Appendix G.
> **In conclusion, given that this assumption holds in many real-world scenarios and can be validated using sensitivity and correlation analyses, we believe it should not be considered as an "unrealistic" assumption.** We thank the reviewer again and have included the corresponding discussions in the revised manuscript (lines 208-212).
>
> > **[W2] How DDIV differs from 2SSI [2].**
>
> We thank the reviewer for the insightful suggestion. As stated in their paper, **2SSI assumes that the unmeasured confounder is conditionally independent of $S$, i.e., $U \not \rightarrow S$.** Therefore, it only addresses two INDEPENDENT confounding and collider biases that coexist simultaneously and **cannot address coupled biases, where $U \rightarrow S$.** We thank the reviewer again and have included a relevant discussion on this point in Appendix A of the revised manuscript.

---

> ### Author Response · Authors · 2025-12-03
>
> > **[W3] Experiments on more real-world datasets.**
>
> We thank the reviewer for the valuable suggestion. Following this suggestion, **we have conducted additional experiments on three more datasets:**
>
> (1) **The semi-synthetic dataset IHDP, which evaluates the performance of DDIV for CATE estimation.** We generated a $3$-dimensional IV vector $\{D_1, D_2, D_3\}$ following a multivariate normal distribution: $\{D_1, D_2, D_3\} \sim \mathcal{N}(\mathbf{0}, \mathbf{I})$. The data generation process for $Y$ and $T$ followed [3], and we introduced selection bias into it through $S \sim \text{Bernoulli}\left(\frac{1}{1 + \exp\left(Y + T + 0.5 * D_3 + 0.2 * D_1 + 0.1 * \left(\sum X_i + \sum U_i\right)\right)}\right)$. Following [4], we report the mean and standard deviation of $\sqrt{PEHE}$, where smaller values indicate better performance.
> | Method    | $\sqrt{PEHE}$  |
> | --------- | --------------------- |
> | Heckit    | 3.723$\pm$0.450       |
> | 2SRI      | 3.149$\pm$0.344       |
> | IPSW      | 2.204$\pm$0.055       |
> | DeepIV    | 2.105$\pm$0.403       |
> | SIV       | 4.039$\pm$0.010       |
> | Kernel IV | 3.512$\pm$0.214       |
> | DeepGMM   | 2.846$\pm$0.464       |
> | DFIV      | 3.713$\pm$0.240       |
> | CBIV      | 3.203$\pm$0.010       |
> | 2SSI      | 2.871$\pm$0.345       |
> | **DDIV**      | **1.411$\pm$0.141**       |
>
> (2, 3) **The real-world SMOKE dataset [1] and CARD dataset [5], which evaluate the performance of DDIV for ATE estimation in more diverse real-world scenarios.** Similar to the Wage2 dataset, we introduced selection bias by applying a non-random sampling strategy to select a subset of the original data. Specifically, we set $S=1$ for those with higher values of SIV and $T$ and lower values of $Y$. We use the results from the previous study [5, 6] as benchmark results and compare the estimated ATEs with them to evaluate the accuracy of ATE estimation, i.e., methods with results closer to the benchmark results perform better.
>
> Results on the SMOKE dataset:
>
> |  Method    | ATE$\pm$95\%CI |
> | --------- | ---------------- |
> | Heckit    | 0.003$\pm$0.001  |
> | 2SRI      | 0.011$\pm$0.005  |
> | IPSW      | -0.116$\pm$0.052 |
> | DeepIV    | 0.000$\pm$0.001  |
> | SIV       | -0.079$\pm$0.038 |
> | Kernel IV | 0.117$\pm$0.54   |
> | DeepGMM   | 0.002$\pm$0.001  |
> | DFIV      | -0.238$\pm$0.062 |
> | CBIV      | -0.013$\pm$0.110 |
> | 2SSI      | -0.087$\pm$0.035 |
> | **DDIV**      | **-0.025$\pm$0.016** |
> | Reference | -0.042$\pm$0.026 |
>
> Results on the CARD dataset:
>
> |  Method   | ATE$\pm$95\%CI |
> | --------- | ---------------- |
> | Heckit | 0.002$\pm$0.001  |
> | 2SRI | 0.007$\pm$0.003  |
> | IPSW | -0.016$\pm$0.007 |
> | DeepIV | 0.006$\pm$0.001  |
> | SIV | 0.024$\pm$0.010  |
> | Kernel IV | -0.351$\pm$0.114|
> | DeepGMM   | -0.003$\pm$0.001|
> | DFIV |0.529$\pm$0.094|
> | CBIV | 0.643$\pm$0.510|
> |2SSI|0.167$\pm$0.105|
> |**DDIV**|**0.091$\pm$0.036**|
> |Reference|0.117$\pm$0.047|
>
>
> **From the results, it is evident that DDIV outperforms all baselines and achieves optimal performance across all three datasets, demonstrating its effectiveness in addressing various real-world challenges.**
>
> ***
> **We hope the above discussion will fully address the reviewer's concerns about our work, and we would really appreciate it if the AC could consider this response and recommend its acceptance for publication. We would like to express our gratitude to both the reviewer and the AC for their great efforts and constructive comments on our manuscript. Thank you!**
>
> > **References**
>
> [1] Wooldridge, J. M. (2016). Introductory econometrics a modern approach. South-Western cengage learning.
>
> [2] Li, B., Wu, A., Xiong, R., & Kuang, K. (2024, July). Two-stage shadow inclusion estimation: an IV approach for causal inference under latent confounding and collider bias. In Forty-first International Conference on Machine Learning.
>
> [3] Wu, A., Kuang, K., Li, B., & Wu, F. (2022, June). Instrumental variable regression with confounder balancing. In International Conference on Machine Learning (pp. 24056-24075). PMLR.
>
> [4] Shalit, U., Johansson, F. D., & Sontag, D. (2017, July). Estimating individual treatment effect: generalization bounds and algorithms. In International conference on machine learning (pp. 3076-3085). PMLR.
>
> [5] Card, D. (1993). Using geographic variation in college proximity to estimate the return to schooling.

---

### Official Review · Reviewer_azTJ · 2025-11-03

**Soundness:** 3
**Presentation:** 3
**Contribution:** 2
**Rating:** 4
**Confidence:** 3

**Summary:**

This paper studied the problem of causal effect estimation under coupled confounding and collider biases. Under this framework, the paper proposed a new identification method for treatment effects under coupled biases with an IV set. The proposed estimation method DualDebiasIV (DDIV) aims to decompose the IV set to separately obtain the SIV and IV, then use them for debiasing and decoupling. Theoretically, the paper showed the correctness of the decomposition and the consistency of the estimate of DDIV. Empirical results further supported the effectiveness of the proposed method.

**Strengths:**

- Overall I found this paper presented an interesting framework for causal effect identification with coupled confounding and collider biases. Such a scenario can be common in practice and the proposed method brings useful insights for practitioners.
- the proposed DDIV algorithm uses a two-stage solution to first identify IV and SIV variables, then adjust for the two biases to estimate the causal effect. This process is very intuitive and natural to apply.
- Theoretically, it is shown that DDIV achieves unbiased estimation. Emprically, the experments with synthetica and real-world data further supported that result.

**Weaknesses:**

My main concerns are about the technical novelty and contribution:
- as the authors claimed, there has been few prior work studying the scenario where the two types of biases are coupled. However, the main technical difficulty seems to stem from the usage of mutual information to identify the IV and SIV sets. Once such sets are obtained, adjusting two biases via regression has been a well-studied solution. Therefore I remain some concern about the technical contribution from this work.

**Questions:**

what is the time complexity for DDLV? is it optimal?

---

> ### Author Response · Authors · 2025-12-03
>
> We sincerely appreciate the reviewer’s great efforts and insightful comments to improve our manuscript. In below, we address these concerns point by point.
>
> > **[W1] The technical contribution from this work.**
>
> We thank the reviewer for the thoughtful comments. However, we respectfully disagree with the statement that "once such sets are obtained, adjusting two biases via regression has been a well-studied solution." **To the best of our knowledge, there is currently no research on the identification and estimation of causal effects under coupled biases.** Even with known IV and SIV sets, existing theories and methods cannot effectively adjust for the coupled two biases. **Therefore, the proposed solution is definitely not "well-studied".**
>
> Our proposed method, built on the IV two-stage regression, introduces the following important technical contributions: (1) in the first stage, we perform decomposed representation learning to learn representations that satisfy IV and SIV conditions, respectively; (2) between the first and second stages, we introduce selection score estimation based on the learned SIV representation; (3) in the second stage, we apply a re-weighting method based on the selection score. **These contributions are original, novel, and also the first approach to treatment effect estimation under coupled biases.**
>
> **Finally, we would like to emphasize our theoretical and technical contributions: we are the first to propose an identification theory for treatment effects under coupled biases, and the first to propose a method for treatment effect estimation under coupled biases.**
>
> > **[Q1] The time complexity for DDIV.**
>
> We thank the reviewer for the valuable suggestion. **We have included a computational complexity analysis in Appendix H of the revised manuscript.** Let $p_{\Phi_t}$, $p_{\Phi_s}$, $p_\mathbf{x}$, $p_{h_1}$, $p_{h_2}$, and $p_{g}$ denote the dimensions of $\Phi_t$, $\Phi_s$, $\mathbf{X}$, $h_1$, $h_2$, and $g$, respectively. $e_{\mathrm{CLUB}}$ denotes the number of CLUB iterations used for MI estimation. **Briefly, the DDIV consists of three sequential procedures: (1) Decomposed representation learning of Stage 1, which has a time complexity of $O(n\cdot(p_\mathbf{x}+  e_{\mathrm{CLUB}}\cdot p_{\Phi_t}\cdot p_{\Phi_s}))$ per iteration; (2) selection score estimation of Stage 1, with a time complexity of $O(n\cdot(p_{\Phi_s} \cdot (p_{h_1}+p_{h_2})+p_\mathbf{x}\cdot (p_{h_1}+p_{h_2}+p_g)))$ per iteration; (3) Stage 2 has a time complexity of $O(n_1 \cdot (p_\mathbf{x}+p_{\Phi_s}))$ per iteration.** The detailed analysis can be found in Appendix H.
>
> Compared to other IV methods based on representation learning (e.g., DFIV with per-iteration time complexity of $O(n \cdot (p_{\mathbf{x}}+p_{\Phi_t}))$ for Stage 1 and $O(n_1 \cdot p_{\mathbf{x}})$ for Stage 2), DDIV has a slightly higher time complexity. This increase is primarily due to: (1) the mutual information estimation in the first stage, which adds $O(e_{\mathrm{CLUB}}\cdot p_{\Phi_t}\cdot p_{\Phi_s})$; (2) the selection score estimation between the first and second stages, which adds $O(n\cdot(p_{\Phi_s} \cdot (p_{h_1}+p_{h_2})+p_\mathbf{x}\cdot (p_{h_1}+p_{h_2}+p_g)))$.
>
> We also provide a runtime comparison of each method on Demand, as shown in the table below.
>
> | Method | Runtime (s) |
> | :-: | :-: |
> | Heckit | 0.10 |
> | 2SRI | 0.11 |
> | IPSW | 0.08 |
> | DeepIV | 108. |
> | SIV | 6.23 |
> | Kernel IV | 8.49 |
> | DeepGMM | 179. |
> | DFIV | 5.31 |
> | CBIV | 1.71 |
> | 2SSI | 6.85 |
> | DDIV | 8.44 |
>
> As shown, while DDIV has a longer running time compared to other methods based on representation learning, such as DFIV and 2SSI, the difference is not significant. Moreover, it is still significantly faster than methods like DeepIV and DeepGMM. **Therefore, although the time complexity of DDIV is not optimal, it is still considered acceptable.**
>
> ***
> **We hope the above discussion will fully address the reviewer's concerns about our work, and we would really appreciate it if the AC could consider this response and recommend its acceptance for publication. We would like to express our gratitude to both the reviewer and the AC for their great efforts and constructive comments on our manuscript. Thank you!**

---

### Meta-Review · Area_Chair_gpVC · 2026-01-08

**Summary:**

This paper introduces a new identification theory and a method, DualDebiasIV (DDIV), that decomposes an IV set to jointly correct both biases.

- novelty concerns: Once IV/SIV are identified (via MI), the rest looks like standard regression-style debiasing; also prior work (2SSI) already tackles joint bias correction

- Assumption of known IV set with disjoint IV and SIV subsets is unrealistic; Strength/credibility and testability of assumptions is of general concern

- Limited real-world validation, artificially injected selection bias

- hand wavy asymptotic results

- Mutual-information penalties may be unreliable

**Reviewer Concerns:**

outstanding concerns:

- Practicality of Assumption 3.1 and how to obtain a valid IV set and split it into IV vs SIV in real data remain unclear

- Positioning vs prior joint-bias work (esp. 2SSI) needs to be toned down.

- Real-world validation remains limited since selection is still artificially introduced

- asymptotic normality and CI validity with learned representations needs to be formalized and grounded.

**Reviewer Scores:**

The reviewers are generally negative, and I doubt the discussion would significantly change the situation.

---

### Decision · Program_Chairs · 2026-01-26

Reject